# Generative Calibration for In-context Learning

**Zhongtao Jiang**[1,2], **Yuanzhe Zhang**[1,2], **Cao Liu**[3], **Jun Zhao**[1,2], **Kang Liu**[1,2,4]

[1]The Laboratory of Cognition and Decision Intelligence for Complex Systems,
Institute of Automation, Chinese Academy of Sciences

[2]School of Artificial Intelligence, University of Chinese Academy of Sciences

[3]Meituan, and [4]Shanghai Artificial Intelligence Laboratory

{zhongtao.jiang, yzzhang, jzhao, kliu}@nlpr.ia.ac.cn, liucao@meituan.com

## Abstract

As one of the most exciting features of large language models (LLMs), in-context learning is a mixed blessing. While it allows users to fast-prototype a task solver with only a few training examples, the performance is generally sensitive to various configurations of the prompt such as the choice or order of the training examples. In this paper, we for the first time theoretically and empirically identify that such a paradox is mainly due to the label shift of the in-context model to the data distribution, in which LLMs shift the label marginal $p(y)$ while having a good label conditional $p(x|y)$. With this understanding, we can simply calibrate the in-context predictive distribution by adjusting the label marginal, which is estimated via Monte-Carlo sampling over the in-context model, i.e., generation of LLMs. We call our approach as *generative calibration*. We conduct exhaustive experiments with 12 text classification tasks and 12 LLMs scaling from 774M to 33B, generally find that the proposed method greatly and consistently outperforms the ICL as well as state-of-the-art calibration methods, by up to 27% absolute in macro-F1. Meanwhile, the proposed method is also stable under different prompt configurations. [1]

## 1 Introduction

The learning paradigm has revolutionized in the era of large language models (LLMs), in which one of the most exciting is in-context learning (ICL) (Brown et al., 2020). Compared to regular supervised learning, LLMs can learn implicitly by prompting a few training examples as demonstrations, i.e., in context. This paradigm allows users including amateurs to fast-prototype a task solver with much less annotated data, while sometimes also gaining remarkable performances.

There are plenty of studies empirically analyzing ICL (Zhao et al., 2021; Lu et al., 2021; Min

et al., 2022; Kim et al., 2022; Wei et al., 2023). Notably, ICL is found to be pathogenically sensitive to the prompt configuration, e.g., the template, choice, and even the order permutation of the training examples can cause the performance to vary from nearly chance to state-of-the-art (Gao et al., 2020; Zhao et al., 2021; Lu et al., 2021). This instability motivates lots of efforts on searching for a better prompt in terms of better few-shot training examples (Rubin et al., 2021; Liu et al., 2021; Su et al., 2022; Wu et al., 2022; Wang et al., 2023), better order permutation of those examples (Lu et al., 2021; Wu et al., 2022), and slot position in the template (Holtzman et al., 2021; Min et al., 2021). Another promising direction is calibration (Zhao et al., 2021; Han et al., 2022; Fei et al., 2023) that adjusts the ICL predictive distribution via an estimated bias. Compared with prompt optimization methods, such a stream is usually much more lightweight and gains substantial and consistency improvement, while also reducing the instability. However, many of those ideas are based on heuristic intuitions, and there are few principled investigations characterizing how the prompt affects the predictive distribution.

This paper fills this gap. In specific, we focus on the text classification task. We study the in-context model $p(x, y)$, i.e., the joint distribution of the input $x$ and label $y$ given a prompt consisting of a few training examples, while the true data distribution is denoted as $q(x, y)$. Specifically, we identify that the in-context model $p(x, y)$ poses label shift (Saerens et al., 2002) to the data distribution $q(x, y)$ both theoretically and empirically. Our theoretical analysis (Section 3.1) is based on the Bayesian interpretation of ICL (Xie et al., 2021; Wang et al., 2023), showing that the in-context model $p(x, y)$ is not ensured to be a valid estimate of the true data distribution $q(x, y)$, due to the prior preference of LLMs and the limited amount of training examples. Next, we empirically find that the in-context label

---

[1]Code implementation is available at https://github.com/changmenseng/generative_calibration.

conditional $p(x|y)$ is a good approximation of the data label conditional $q(x|y)$, while the in-context label marginal $p(y)$ generally deviates from the true one $q(y)$, which provides persuasive evidence of label shift in the in-context model (Section 3.2). Then, with this understanding, we can calibrate the in-context classifier $p(y|x)$ by simply adjusting the label marginal $p(y)$, which is naturally obtained by marginalizing out the input $x$ and can be effectively estimated via Monte-Carlo sampling, i.e., generating new instances from the in-context model. We call our approach generative calibration (GC).

Actually, most previous state-of-the-art calibration approaches (Zhao et al., 2021; Han et al., 2022; Fei et al., 2023) implicitly assume that the ICL predictive distribution has a shift on label marginal. However, none of them validate the soundness of that assumption, while we for the first time verify this systematically. What's more, their estimates of the label marginal are too heuristic to be solid, while our Monte-Carlo estimate is an unbiased one to the in-context label marginal $p(y)$.

We conduct exhaustive experiments on 12 text classification tasks and 12 LLMs scaling from 774M to 33B. The results show that the proposed GC greatly and consistently improves the performance of ICL (by up to 27% absolute in macro-F1) and outperforms previous state-of-the-art calibration methods (by up to 9% absolute in macro-F1) for all LLM scales. Meanwhile, GC is also stable towards changes of the choice and order of training examples in the prompt, and exceeds or be competitive to prompt optimization methods, even though some of which are not in the true few-shot learning setting (Perez et al., 2021). Overall, GC is a lightweight and effective approach making LLMs better few-shot learners and saves the effort of cumbersome prompt engineering.

## 2 Background

In the ICL paradigm, we have a few training examples $\mathcal{D}_t = \{e_i\}_{i=1}^K$, where each one is independently and identically sampled from the data distribution, which we denote as $q(e)$. Prompting a language model LM with those examples, the continuation distribution is a generative model:

$$p(e|\mathcal{D}_t^\pi) = p_{\text{LM}}(\mathcal{T}(e)|\mathcal{D}(\mathcal{D}_t^\pi))$$
$$\mathcal{D}(\mathcal{D}_t^\pi) = \oplus_{i=1}^K \mathcal{T}(e_{\pi(i)}) \tag{1}$$

where $\pi$ represents a specific order permutation, $\oplus$ denotes the text concatenation operation, $\mathcal{T}(\cdot)$ uses

a template to format the example to be a demonstration. We call this distribution the *in-context model* and use $p(e)$ as its shorthand: $p(e) := p(e|\mathcal{D}_t^\pi)$.

In this work, ICL is utilized for classification tasks. Then, each training example $e_i$ is a tuple: $e_i = (x_i, y_i)$, where $x \in \mathcal{X}$ is the input sequence and $y \in \mathcal{Y}$ is its corresponding label. In this case, the generative model factorizes as:

$$p(x,y) = p(x)p(y|x) =$$
$$p_{\text{LM}}(\mathcal{T}(x)|\mathcal{D}(\mathcal{D}_t^\pi)) p_{\text{LM}}(\mathcal{T}(y)|\mathcal{D}(\mathcal{D}_t^\pi) \oplus \mathcal{T}(x)) \tag{2}$$

In most cases, we only use the classifier $p(y|x)$ to predict the label of the given input sequence $x$. By this intuitive construction, the classifier performs surprisingly well on many NLP tasks, sometimes even achieving state-of-the-art.

## 3 Distribution Shift of In-context Model

In this section, we provide both theoretical and empirical evidence showing that the in-context model has a distribution shift on the label.

### 3.1 A Bayesian View

It is shown that the in-context model would imitate the prompting examples to generate similar sequences (Meyerson et al., 2023). One principled and popular explanation is that such a model is an approximation of the posterior predictive distribution in Bayesian statistics (Xie et al., 2021; Wang et al., 2023):

$$p(x,y|\mathcal{D}_t^\pi) \approx \int p(\theta|\mathcal{D}_t)p(x,y|\theta)d\theta$$
$$p(\theta|\mathcal{D}_t) = \frac{p(\theta)\prod_{i=1}^K p(x_i,y_i|\theta)}{p(\mathcal{D}_t)} \tag{3}$$

where $\theta \in \Theta$ is a latent parameter controlling the "topic" of the sequence like LDA (Pritchard et al., 2000) topic model[2]. $p(\theta|\mathcal{D}_t)$ is the posterior given the training examples, which softly selects the topic. Also note that this identity approximately holds ($\approx$), since Bayesian approaches consider all the examples to be exchangeable[3], while clearly, LLMs have order preference. We assume that LLMs have

---

[2]The topic could be viewed as the task.

[3]Exchangeability means that the joint distribution of a sequence $e_{1:K}$ is invariant to any order permutations $p(x_{1:K}) = p(x_{\pi(1):\pi(K)})$. According to de Finetti's theorem (Aldous et al., 1985), if a random sequence is exchangeable, it is then equivalent to a mixture model that each sample is conditional identically independent given a latent parameter, which forms the null hypothesis of Bayesian inference.

a topic supporting the data distribution of interest, i.e., there exists $\theta^\star \in \Theta$ such that $p(\theta^\star) > 0$ and $p(x, y|\theta^\star) = q(x, y)$. Ideally, if the posterior recognizes the true topic $\theta^\star$ of the training examples without uncertainty, i.e., $p(\theta|\mathcal{D}_t) = \delta(\theta - \theta^\star)$, then the in-context model $p(x, y)$ approaches the true data distribution $q(x, y)$ exactly. According to Schwartz's theorem for posterior consistency (Schwartz, 1965), this ideal case can be asymptotically realized as the number of training examples $K$ goes to infinity, which is impossible for LLMs[4].

When only a few training examples are acceptable to LLMs, there are at least two negative consequences. On one hand, as shown in Equation (3), the prior preference of LLMs, i.e., topic prior $p(\theta)$, would dominate the posterior $p(\theta|\mathcal{D}_t)$. Empirical evidence include common token bias (Zhao et al., 2021; Razeghi et al., 2022) and prediction insensitivity when the training labels are corrupted when $K$ is small (Min et al., 2022; Wei et al., 2023; Pan et al., 2023). On the other hand, it is not enough to express the data distribution with just a few training examples. To see this, consider a movie review sentiment classification task. Suppose there exists a topic $\theta'$ such that $p(x, y|\theta')$ only generates positive reviews[5]. When we happen to have only a few positive samples in hand, the population of the posterior $p(\theta|\mathcal{D}_t)$ is easy to fall into regions close to $\theta'$ instead of the true topic $\theta^\star$, which would shift the label marginal $p(x|y)$ so that the probability of positive label is risen to a level far beyond the true one. While being widely-observed in previous works in terms of majority bias (Zhao et al., 2021) and underspecification (Si et al., 2023), this phenomenon is formally termed as *label shift* (Schölkopf et al., 2012), which proposes that 1) the model and data share the same label conditional: $p(x|y) = q(x|y)$, and 2) differ in the label marginal: $p(y) \neq q(y)$.

Besides this theoretical analysis, in what follows, we empirically validate that label shift exists in in-context models.

## 3.2 Label Shift Empirical Validation

By assuming to be accessible to a labeled validation set[6], we now empirically verify two points

of label shift: $p(x|y) = q(x|y)$ and $p(y) \neq q(y)$. While the second point is straightforward by simply comparing the empirical label marginals, the first one poses a challenge given that we are agnostic to the true model $q(x, y)$ but only some samples from it (i.e., the validation set). As a surrogate, we don't seek to verify $p(x|y)$ approaches $q(x|y)$ exactly. We shall show that $p(x|y)$ performs very well in ranking the examples, illustrating that it is a good approximation of $q(x|y)$. Then, our goal is to verify: 1) in-context label conditional $p(x|y)$ is good to $q(x|y)$ and 2) in-context label marginal $p(y)$ is different from the data label marginal $q(y)$.

Our investigation is on SST2 dataset (Socher et al., 2013) using GPT2-XL (1.5B) (Radford et al., 2019), GPT-NEO (2.7B) (Black et al., 2021) and GPT-J (6B) (Wang, 2021; Wang and Komatsuzaki, 2021). For simplicity, we only display results of GPT2-XL (1.5B) and left others in Appendix E.

### 3.2.1 In-context Label Conditional is Good

Without the label marginal $p(y)$, the label conditional $p(x|y)$ alone can't tell if the input $x$ belongs to the class $y$. However, it can tell which of the two samples $x$ and $x'$ is more likely to belong to $y$ instead of other classes. In specific, let's first consider a binary classification task in which $Y = \{N, P\}$, where we denote $N$ and $P$ as negative and positive, respectively. We score each input $x$ as the ratio of label conditionals:

$$s(x) := \frac{p(x|P)}{p(x|N)} = \frac{p(N)p(P|x)}{p(P)p(N|x)} \propto \frac{p(P|x)}{p(N|x)} \quad (4)$$

What does this score mean? Since the score is proportional to the odds: the ratio of the probability that the in-context recognition model $p(y|x)$ predicts positive to the probability that it doesn't, it measures how much more confidence of the positive prediction than the negative prediction. Given this score, we can obtain a rank $r_s$ of the entire validation set, where examples of the higher ranking are more likely to be positive. The rank quality can be justified by the receiver operating characteristic curve (ROC) and the area under it AUROC[7]. AUROC formally stands for the empirical probability that a random positive sample is ranked above a negative sample (Zhou, 2021). Concretely, denote the validation set $\mathcal{D}_v = \mathcal{D}_v^P \cup \mathcal{D}_v^N$, where the superscript represents the subset that contains all the

---

[4]Though there may also exists other conditions for a good posterior estimation, e.g., high-quality training examples, we just can't ensure them to happen in the few-shot scenario.

[5]It is reasonable to believe that $\theta'$ has support in the topic prior such that $p(\theta') > 0$, since there are contiguous positive reviews in the pretraining corpus.

[6]Validation set violates the true few-shot learning setting only when it's used for model development or selection. It is

hard to analyze the model if we don't have a validation set for grounding results.

[7]In multiple-class case, we use macro one-to-one AUROC, which is abbreviated as macro-AUROC for simplicity.

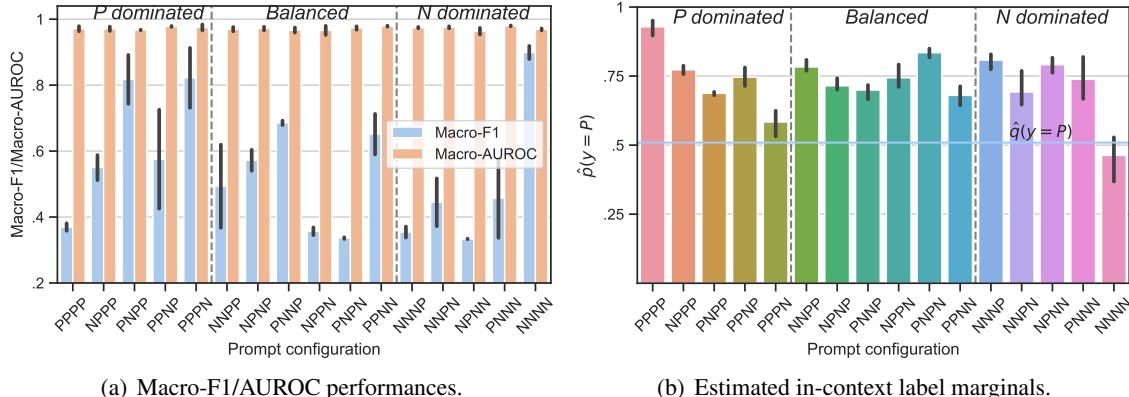

(a) Macro-F1/AUROC performances.

(b) Estimated in-context label marginals.

Figure 1: Macro-F1/AUROC performances (a) and estimated in-context label marginals (b) of GPT2-XL (1.5B) across different prompt configurations on SST2.

examples in the corresponding label, AUROC is:

$$\frac{\sum_{x^P \in \mathcal{D}_v^P} \sum_{x^N \in \mathcal{D}_v^N} 1(r_s(x^P) > r_s(x^N))}{|\mathcal{D}_v^P||\mathcal{D}_v^N|} \quad (5)$$

where $r_s(x)$ is the rank of $x$. Now, if a model is perfect, i.e., $p(x|y) = q(x|y)$, it will place all the positive examples above the negative examples, where AUROC reaches its maximum 1. Therefore AUROC somehow measures the closeness between $p(x|y)$ and $q(x|y)$.

Since the rank is invariant to the $x$-independent term, e.g., $p(P)/p(N)$ in Equation (4), we can use the odds of the classification distribution $p(y|x)$ to compute AUROC, while this metric is actually evaluating $p(x|y)$. We evaluate the 4-shot ICL performance with prompts having different class balances and orders on SST2, where each prompt configuration is evaluated in three random runs. The results in F1 and AUROC of GPT2-XL (1.5B) are shown in Figure 1(a), where the x-axis represents different prompt configurations. For example, "PNPP" indicates three positive examples and one negative example ordered in the second place in the prompt.

We can obtain two findings: 1) On average, ICL achieves very high AUROC values that don't match the F1 across prompt configurations: the average AUROC performance typically exceeds 0.95, while the average F1 performance hardly reaches 0.8. 2) In contrast to the sensitivity of F1, AUROC is stable to the prompt changes with low variances. These findings provide strong evidence that $p(x|y)$ is a stable good approximation of $q(x|y)$ no matter of prompt configurations, roughly establishing $p(x|y) \approx q(x|y)$.

### 3.2.2 In-context Label Marginal is Different

The second point, i.e., $p(y) \neq q(y)$ is straightforward to verify by comparing the empirical label marginals of the in-context model and data. First, we need to estimate the in-context label marginal, which is the marginalization of the joint distribution:

$$p(y) = \sum_{x \in \mathcal{X}} p(x)p(y|x) \quad (6)$$

The exact marginalization is intractable since we can't enumerate all the sequences in the input space $\mathcal{X}$. Therefore, we apply Monte-Carlo sampling to obtain an unbiased estimate:

$$p(y) \simeq \hat{p}(y)$$
$$= \frac{1}{L} \sum_{l=1}^{L} p_{\text{LM}} \left( \mathscr{T}(y) | \mathscr{D}(\mathcal{D}_t^\pi) \oplus \mathscr{T}(x^l) \right) \quad (7)$$

where $x^l$ is a sample from $p_{\text{LM}}(\mathscr{T}(x)|\mathscr{D}(\mathcal{D}_t^\pi))$, i.e., a generated sequence of LLMs prompted with training examples in $\mathcal{D}_t$. The whole process is shown in Figure 2. In this paper, we set $L = 100$, which is enough for a stable estimation (Details are shown in Appendix J). As for the data label marginal $q(y)$, simply counting the number of different labels in the validation set and then normalizing them forms a maximized likelihood estimate. We plot the estimated in-context model and data marginal probability of the positive label $\hat{p}(y = P)$ and $\hat{q}(y = P)$ in Figure 1(b). We can see that the in-context label marginal deviates from the data label marginal in most cases. Concretely, the deviation extent positively correlated with the majority label, which is in expectation as discussed in Section 3.1. Also, GPT2-XL (1.5B) has much prior

Figure 2: Illustration of the marginal distribution estimation on SST2. We first sample $L$ sequences (the generation ends when it meets line break "\n") conditioned on the prompt concatenated with few-shot training examples and the input prompt phrase ("\nReview:" in this case), and next concatenate the output prompt phrase ("\nSentiment:" in this case) to obtain their in-context predictive distributions. The in-context label marginal is then estimated by averaging those in-context predictive distributions.

preferences on the positive label, where except the case when all the training examples are negative, i.e., "NNNN", the marginal probability of the label positive exceeds 0.5 in other cases.

In short summary, in-context models are mainly biased by shifting the label marginal, which is generally implicitly assumed in previous works (Zhao et al., 2021; Han et al., 2022; Fei et al., 2023). However, to the best of our knowledge, we are the first to verify this systematically.

## 4 Generative Calibration

With the understanding in the previous section, if we accept that $p(x|y) = q(x|y)$, the solution is quite straightforward, that is, adjusting the label marginal to the desired one yields the true label predictive distribution:

$$q(y|x) = \frac{q(y)q(x|y)}{q(x)} \propto \frac{q(y)}{p(y)}p(y|x) \qquad (8)$$

where the model label marginal $p(y)$ is estimated via Equation (7). The data label marginal $q(y)$ is hard to estimate since we only have a few training examples in our setting. Previous works (Zhao et al., 2021; Min et al., 2021; Han et al., 2022; Fei et al., 2023) typically assume $q(y)$ to be uniform, yielding the following classifier:

$$\tilde{q}(y|x) \propto \frac{p(y|x)}{\tilde{p}(y)} \qquad (9)$$

We follow this convention and leave accurate estimation of data label marginal $q(y)$ in future works[8].

Since our method involves multiple generations (for estimating $p(y)$), we denote it as generative calibration (GC). GC could be also explained via cost-sensitive learning theory (Elkan, 2001; Ling and Sheng, 2008), where the label marginal is one of the most widely-used costs (Buda et al., 2018).

Although previous state-of-the-art calibration methods (Zhao et al., 2021; Han et al., 2022; Fei et al., 2023) share the same form as ours in Equation (9), in contrast to our principled unbiased estimate, their estimates are too heuristic to be solid. For example, contextual calibration (Zhao et al., 2021) estimates the label marginal via heuristically constructed seemingly context-free texts such as "N/A": $\hat{p}(y) = p(y|"N/A")$. However, these texts are never verified to be context-free as claimed, because LLMs might have a preference bias on them. Also, LLMs barely see these texts in the pretraining corpus, which would pose an out-of-distribution (OOD) (Kim et al., 2020) problem.

## 5 Experiments

### 5.1 Setups

**Datasets**

We use 12 text classification tasks in our experiments: SST2 and SST5 (Socher et al., 2013), CR (Hu and Liu, 2004), MR (Pang and Lee, 2005), SUBJ (Pang and Lee, 2004), AGNews (Zhang

---

[8]Some methods to estimate the data label marginal include BBSE (Lipton et al., 2018; Azizzadenesheli et al., 2019) or

EM algorithm in Saerens et al. (2002). Both methods involve estimating expected statistics of $p(x, y)$, which is done by generations similar to estimating $p(y)$. However, we find that their estimates are extremely unstable in our case, causing severe performance drops even compared with vanilla ICL.

et al., 2015), DBPedia (Zhang et al., 2015), TREC (Voorhees and Tice, 2000), CB (De Marneffe et al., 2019), RTE (Dagan et al., 2006), QQP (DataCanary et al., 2017) and SNLI (Bowman et al., 2015). For datasets whose testing set size is greater than 2000, we sub-sample 2000 examples for evaluation. The prompt template and example generations of each dataset are shown in Appendix A and B, respectively.

**Language Models**

We investigate 12 LLMs in a wide range of scales, including Transformer-based models GPT2-Large (774M), GPT2-XL (1.5B) (Radford et al., 2019), GPT-NEO (2.7B) (Black et al., 2021), GPT-J (6B) (Wang, 2021; Wang and Komatsuzaki, 2021), GPT-NEOX (20B) (Andonian et al., 2021; Black et al., 2022), OPT (13B and 30B) (Zhang et al., 2022), LLaMA (13B and 33B) (Touvron et al., 2023) and recently proposed RNN-based models RWKV (3B, 7B, and 14B) (Peng et al., 2023).

**Compared Methods**

Besides vanilla ICL, we include the following state-of-the-art ICL calibration methods for comparison: 1) *Noisy channel* (NC) (Min et al., 2021) changes the slot position of the input and output in the template, and then uses the label likelihood $p(x|y) = p_{\text{LM}}(\mathcal{T}(x)|\mathscr{D}^{-1}(\mathcal{D}_t^\pi), \mathcal{T}(y))$ for prediction, where $\mathscr{D}^{-1}(\mathcal{D}_t^\pi)$ denotes the concatenation of flipped demonstrations. Since $p(x|y) \propto p(x|y)/p(y)$, this method actually has the same form as ours in Equation (9), so we categorize it as a calibration method.
2) *Contextual calibration* (CC) (Zhao et al., 2021) estimates the label marginal via context-free texts.
3) *Domain-context calibration* (DC) (Fei et al., 2023) proposes a further requirement for the context-free texts: they must be also context-free in the task domain. They construct such domain context-free texts by randomly sampling and concatenating words from the task dataset.
4) *Prototypical calibration* (PC) (Han et al., 2022) learns a Gaussian mixture model (GMM) from the output probability vectors. They then consider each cluster corresponds uniquely to a label, where the learned cluster weights are the estimated label marginal. For a fair comparison, we learn the GMM on the set of generative sequences as the same as GC.

We consider 2, 4, and 8-shot true few-shot learning settings. For evaluating each method, we randomly sample the original training set of the dataset to construct the training examples. LLMs scaling less than 30B are evaluated in 5 runs, while those larger than 30B are evaluated in 2 runs using different random seeds. This finally yields 1944 runs for each method, which the results should be solid. The performance is measured by macro-F1. Implementation details are shown in Appendix C. We also present time complexity analysis in Appendix D.

## 5.2 Main Results

We plot the average 4-shot performance across different datasets in different runs in Figure 3. For the 2 and 8-shot results please refer to Appendix F. We find that our proposed GC is effective in the following aspects:
1) The proposed GC significantly improves the ICL performance, by up to 27% absolute (2-shot case of GPT2-Large (774M)). The improvement is consistency for LLMs in all parameter scales. Notably, our approach enables small models to outperform larger models' vanilla ICL performances. For example, GC lifts GPT2-XL (1.5B) to 0.64 in macro-F1, while OPT's (30B) ICL performance is only 0.57, despite being over 20 times larger. Also, note that the improvement is more obvious for smaller models, which is useful in the limited resources scenario.
2) The proposed GC outperforms all the calibration baselines, by up to 7% absolute (4-shot case of LLaMA (33B)), suggesting that our method is not only theoretically guaranteed to be superior (being an unbiased estimate of $p(y)$), but also empirically verified to be better.

## 6 Analysis

Following the finding that GC generally outperforms ICL and other calibration methods, we conduct further analysis in this section.

### 6.1 Robustness

Since we've shown in Section 3 that $p(x|y)$ is a good and stable approximation of $q(x|y)$ no matter of the prompt configurations, if the estimated $p(y)$ is solid, then GC is expected to be also robust towards changes of the prompt. This is verified empirically, as shown in Figure 4 that depicts the 4-shot performance distribution of 10 randomly sampled training sets and 10 random order arrangements of one particular training set for GPT2-XL

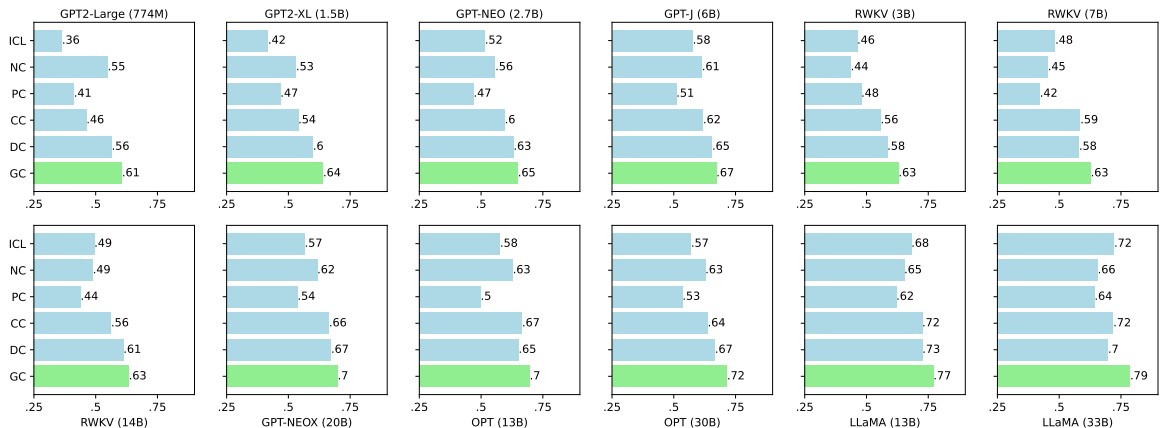

Figure 3: Average 4-shot performances on 12 datasets, where the best result of each LLM is colored green. We also conduct Hotelling's t-square test (Hotelling, 1992) to show that the improvement of GC is statistically significant in the level of significance 0.01, as shown in Appendix G.

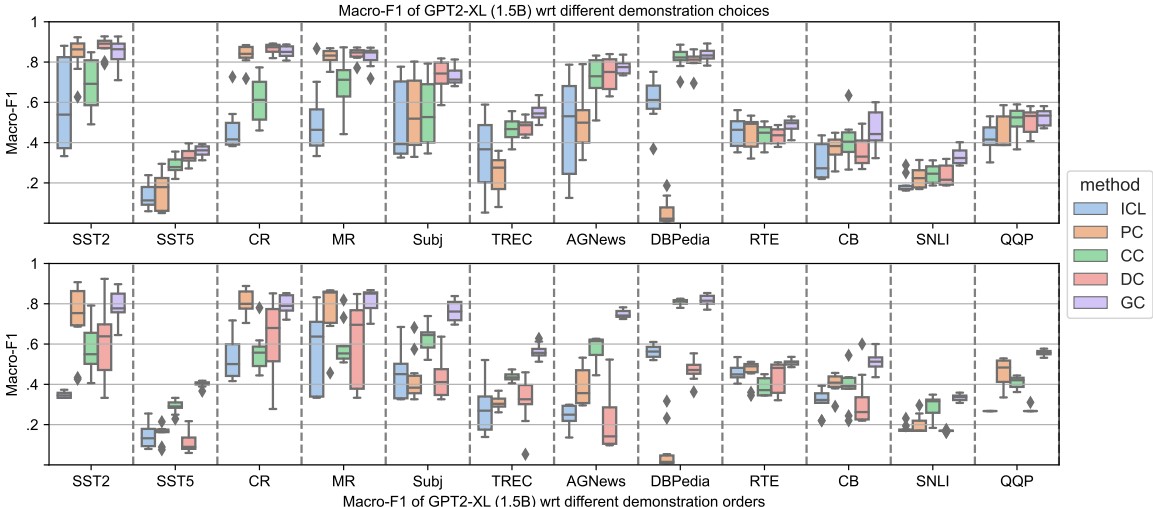

Figure 4: Sensitivity results for GPT2-XL (1.5B).

(1.5B) (Results of GPT-NEO (2.7B), GPT-J (6B), and LLaMA (13B) can be found in Appendix H). It is obvious that the proposed GC greatly and consistently reduces the performance variance for almost all datasets. For instance, GC reduces the choice standard deviation from 0.25 to 0.03 for GPT2-XL (1.5B) on AGNews, while CC only achieves 0.10.

To further show that GC does address the majority and recency bias (Zhao et al., 2021) introduced by the choice and order of training examples, we conduct 4-shot experiments with prompts that have different class balances and orders in SST2 dataset, where each prompt configuration is evaluated in three random runs. The results of GPT-XL (1.5B) are shown in Figure 5. According to Zhao et al. (2021) and our analysis in Section 3.2, LLMs tend to predict labels that appear more frequently in the prompt and are closer to the testing input, which

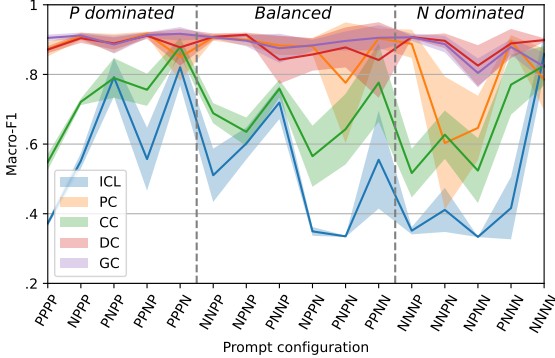

Figure 5: 4-shot results of GPT2-XL (1.5B) on SST-2 using prompts with different configurations.

is damageable to the performance. For example, if all the training examples are positive, i.e., "PPPP" in Figure 5, LLMs would pathogenically predict all the testing inputs as positive, leading to an extremely low F1 performance. As shown, our pro-

posed GC greatly and consistently alleviates this issue across all the prompt configurations, where the increase is up to 54%.

## 6.2 Comparison with Prompt Optimization Methods

In addition to calibration methods, we also compare our approach against prompt optimization methods for order and choice of training examples, where 4-shot results for all LLMs are shown in Figure 6 (2 and 8-shot results are shown in Appendix I).

### 6.2.1 Order

We compare GC with LocalE and GlobalE (Lu et al., 2021), which aim to find the best order permutation of the training examples in the true few-shot setting based on heuristic scores. We only reproduce the 2 and 4-shot results because these methods would rate every order permutation, where the complexity is $\mathcal{O}(K!)$ and larger shot is computational prohibitive. As shown in Figure 6, our proposed GC exceeds both order optimization methods by considerable margins, while being much more efficient with complexity $\mathcal{O}(L)$.

### 6.2.2 Choice

We compare GC with KATE (Liu et al., 2021), which for each testing example, it retrieves the $K$-th most similar training examples in the whole training set to construct the prompt. Note that this method violates the true-few shot learning setting by accessing a large training set containing much more examples than that of ours[9]. We include two implementations that use BM25 and Roberta-Large fine-tuned on text similarity and natural language inference tasks (Reimers and Gurevych, 2019) for retrieval, denoted as KATE-BM25 and KATE-RoBERTa. As shown in Figure 6, though with only a few training examples, the proposed GC is usually competitive with choice optimization methods. For example, 4-shot performance gaps between GC and KATE-BM25 don't exceed 0.03 for GPT2-XL (1.5B), OPT (13B), OPT (30B), LLaMA (13B) and LLaMA (33B).

In general, GC outperforms or is competitive with strong prompt optimization methods, thus saving the effort of cumbersome prompt engineering.

---

[9]For example, SNLI has 56k training examples in the training set, which is over 10k times more than our case.

## 7 Related Works

In-context learning, or few-shot prompting (Brown et al., 2020) has become one of the most curious emergent abilities (Wei et al., 2022a) of LLMs, which benefits in fast-prototyping and much less requirement of annotated data. ICL also forms the basis of its further extensions, including chain of thoughts (CoT) (Wei et al., 2022b; Wang et al., 2022; Lyu et al., 2023), in-context instruction learning (ICIL) (Ye et al., 2023), meta-ICL (Coda-Forno et al., 2023) and so on.

There are many works empirically analyzing in-context learning, which generally find that in-context learning is pathogenically unstable towards the template, choice, and order of the training examples (Gao et al., 2020; Zhao et al., 2021; Lu et al., 2021). This motivates numerous works trying to find a better prompt in terms of template (Min et al., 2021), choice (Rubin et al., 2021; Su et al., 2022; Wang et al., 2023; Iter et al., 2023; An et al., 2023), and order of the training examples (Lu et al., 2021; Wu et al., 2022). However, most of them either violate the true few-shot learning (Perez et al., 2021), or are complicated learning-based (Wang et al., 2023), or both (Rubin et al., 2021), which deviates from the original intention of ICL. On the other hand, except for a few works that fine-tune LLMs to learn in-context (Chen et al., 2021; Coda-Forno et al., 2023), most LLMs are pretrained on a large amount of raw text. So it is not reasonable to believe that LLMs faithfully express the label distribution of a given task. To address this, other lines of work try to calibrate the in-context classifier by estimating and then countering this bias introduced from the prompt (Zhao et al., 2021; Yang et al., 2023; Han et al., 2022). These methods are much more lightweight compared to prompt optimization methods but fail in their heuristic bias estimations.

Theoretical understanding is also attractive to researchers. For example, Dai et al. (2022); von Oswald et al. (2022) find surprising similarities between ICL and gradient descent formally and empirically, establishing explanations via implicit gradient descent. Han et al. (2023) provide an explanation of ICL under the view of kernel regression. Xie et al. (2021) demonstrate ICL as implicit Bayesian inference, where it occurs when the LLM infers a shared latent topic of demonstrations in the prompt. While their theory is appealing, they assume the pre-trained data distribution as a mixture of hidden Markov models (HMMs), and con-

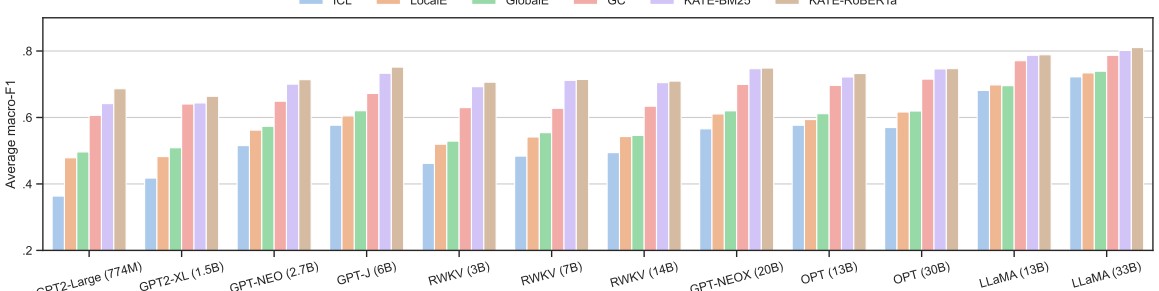

Figure 6: Average 4-shot performance comparison results with prompt optimization methods.

duct experiments on a small-scale synthetic dataset, which is far different from the real case.

## 8 Conclusion

In this work, we for the first time theoretically and empirically identify that the in-context model mainly poses a label shift to the data distribution. We then propose generative calibration, a lightweight solution to mitigate this distribution shift. Exhaustive experimental results on 12 tasks and 12 LLMs demonstrate the effectiveness of our approach, in its improvement and stability.

## Acknowledgement

This work was supported by the National Key R&D Program of China (2022ZD0160503) and the National Natural Science Foundation of China (No.62276264). This research was also supported by Meituan.

## Limitations

The main limitation of this work is to assume the data label marginal $q(y)$ to be uniform. Though to the best of our knowledge there is no work researching this problem and previous works also follow this assumption, this assumption is not true in general realistic cases containing much label imbalance scenarios. However, estimating $q(y)$ in a true few-shot learning setting is challenging since we only have a few training samples and an unlabeled testing set in hand. We've actually experimented with BBSE (Lipton et al., 2018) and EM algorithm in (Saerens et al., 2002), which naturally fit our idea because they require samples from the model distribution. However, the estimate is just plausible in a few runs accidentally, in general we fail to obtain consistent solid estimates, especially for the dataset which the number of labels is large such as DBPedia. Although this limitation is important, it

doesn't affect main claims of this paper. We leave this point for future works.

This work is also limited by using AUROC to measure the closeness of $p(x|y)$ and $q(x|y)$. Note that AUROC is high is just a necessary condition to say $p(x|y)$ and $q(x|y)$ are close, but not the sufficient condition. Therefore, when AURCO is high, we can't rigorously say the $p(x|y)$ and $q(x|y)$ are close. To the matter of fact, for our method to be effective, we don't need the model label conditional $p(x|y)$ to be close to the true one, we only need the in-context model to perform well in inter-label ranking: it can put most of the positive examples in the front, and most of the negative samples in the back (in a binary classification setting). If so, all we need is to set a proper decision threshold (an 1D decision boundary), which is actually what our calibration method do: if the model has shift on the positive label, i.e., $p(P) > p(N)$, the proposed GC actually increases the decision threshold from the default 1 to $\frac{p(P)}{p(N)}$, i.e., predict $P$ when $\frac{p(P|x)}{p(N|x)} > \frac{p(P)}{p(N)}$ and predict $N$ otherwise. Choosing this decision boundary can ensure that the model label marginal is uniform but not biased to any specific labels. So a high AUROC is actually enough for our method to work.

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

## A Prompt Template

Table 1 shows prompt templates used for different datasets.

## B Example Generations

Table 2 shows example generations using GPT2-XL (1.5B) of different datasets. Note that the sequences are generated by vanilla sampling, therefore they might not be a coherent sentence. But they are reasonable to estimate the in-context label marginal.

## C Implementation Details

We use LLM.int8() (Dettmers et al., 2022) quantization for all LLMs to reduce the memory usage[10]. This techniques allows us to use LLMs in a few consumer GPUs. Specifically, we use two servers with 8 NVIDIA GeForce RTX 2080Ti (11GB memory) and 10 NVIDIA GeForce RTX 3090 (24GB memory) in the experiments, where device usages of different LLMs are shown in Tabel 3. We cache the prompt hidden states to improve the inference speed since the ICL prompt remains unchanged for every testing example. Note that RNN-based models RWKV requires much less memory compared to Transformer-based models in the same scale. This is because the cached hidden states of RWKV are only of length 1, while that of Transformer-based models equal to the prompt length.

In the generation process, the prompt is the ICL demonstration prompt $\mathscr{D}$ plus the input slot name, e.g., "Review: $<x>$\nSentiment: $<y>$\n\nReview:"

---

[10]HuggingFace's transformers library supports this feature, see https://huggingface.co/docs/transformers/v4.30.0/en/perf_infer_gpu_one.

| Dataset | Prompt Template | Label Words |
|---------|-----------------|-------------|
| SST2 | Review: $<x>$\nSentiment: $<y>$ | negative, positive |
| SST5 | Review: $<x>$\nSentiment: $<y>$ | terrible, bad, neutral, good, great |
| CR | Review: $<x>$\nSentiment: $<y>$ | negative, positive |
| MR | Review: $<x>$\nSentiment: $<y>$ | negative, positive |
| Subj | Input: $<x>$\nType: $<y>$ | objective, subjective |
| TREC | Question: $<x>$\nAnswer Type: $<y>$ | abbreviation, entity, description, person, location, number |
| AGNews | Input: $<x>$\nType: $<y>$ | world, sports, business, technology |
| DBPedia | Input: $<x>$\nType: $<y>$ | company, school, artist, athlete, politics, transportation, building, nature, village, animal, plant, album, film, book |
| RTE | Premise: $<x^1>$\nHypothesis: $<x^2>$\nPrediction: $<y>$ | true, false |
| CB | Premise: $<x^1>$\nHypothesis: $<x^2>$\nPrediction: $<y>$ | true, false, neither |
| SNLI | Premise: $<x^1>$\nHypothesis: $<x^2>$\nPrediction: $<y>$ | entailment, neutral, contradiction |
| QQP | Question1: $<x^1>$\nQuestion2: $<x^2>$\nPrediction: $<y>$ | false, true |

Table 1: Prompt templates and label words of different datasets, where $x$ and $y$ denote the input sequence and output label, respectively. Each demonstration is split by double line breaks "\n\n".

for SST2. Given that each slot is split by a line break as shown in Table 1, when the generation encounters a line break, we force the next contiguous tokens to be the next slot phrase of the template. The maximum generated length is set to 384.

## D Time Complexity

In deployment, if we have $N$ testing examples, the method includes $L$ generations and $N$ ICL inferences, which the time complexity is $O(L + N)$. When $N \gg L$, the additional time cost of the generation can be neglected. Time complexities of different methods are shown in Table 4. As seen, our method is superior or on par to previous methods.

## E Additional Results of Label Shift Empirical Validation

As a supplement to Section 3.2, Figure 7 and 8 display the macro-F1/AUROC performances and estimated in-context label marginals across different prompt configurations on SST2 of GPT-NEO (2.7B) and GPT-J (6B), respectively. The results show similar trends with that of GPT2-XL (1.5B), as detailed in 3.2.

## F Full Results of Main Experiments

We provide full results of main experiments in section 5. Figure 9 shows the average 2 and 8-shot performances on 12 datasets. We also show 2, 4, and 8-shot full results (averaged by random runs) of different calibration methods on different datasets in Table 5, 6, and 7.

## G Hotelling's T-square test

We show that the improvement of GC to the other methods is statistically significant. Concretely, the complete method performance is represented by a matrix with shape $M \times 12$, where each row contains results of 12 datasets in a random run (each run has different training examples randomly sample from the dataset training set). We then use Hotelling's t-square test (Hotelling, 1992) to test whether the 12-dimensional performance vectors of two methods have the same mean, where the null hypothesis is that their means are the same. To compute the test static, the covariance matrix of samples should be invertible, requiring $M > 12$, while in our main experiments, we only run each method in 5 or 2 random runs. Since running each method of each LLM more than 12 times is prohibitively time-consuming, here we only consider the 4-shot case of GPT2-XL (1.5B), where we set $M = 50$. As shown in Table 8, GC is significant different from other methods given that the p-values are lower than 0.01.

| Dataset | Generated data |
|---|---|
| SST2 | Review: everyone else should watch this; don't be surprised stingy, it might be your last one.
Review: any playthrough starts with a score of 2-3up, best games worth the coin! |
| SST5 | Review: a grim, disgusting, thoroughly pleasant reflection on mankind.
Review: on first viewing it feels like the actors are auditioning to star in a gag reel. |
| CR | Review: It is a wonderful tool and i would use it without worry. one moment to the next.
Review: good photo i put to it all took about 4 hrs |
| MR | Review: it's hard to freeze an action sequence.
Review: he has thus managed to reconcile his proven wavelike power with the sheer inevitable tre future friction frustration of this endless writing process. |
| Subj | Input: can you imagine ever audition meeting anyone else in the world who can bear your burden, miss kumin surely.
Input: for the last 10 weeks, when my 100 BOP brother and I, have been in US, we went to trade mart everyday every day... This is the kind of effect it has!. |
| TREC | Question: What was the official reason given for Ben Franklin's death?
Question: What city is named for San Francisco's Golden Gate Bridge? |
| AGNews | Input: 446 billion times faster Than Neptune 1 June 2018 26:53m The counter-clockwise nucleus of Europa hinted at at experiences better than those quenched by hyperplaces. or ancient enigmasmas
Input: BBC browsers golden age loses its shine The golden age of BBC browsers looks set to date back to 1996, with both IE Microsoft and F1 team Lotus falling prey to penalty points.. |
| DBPedia | Input: Rush Limbaugh is an American radio host, radio talk show host, and raconteur who hosts, produces and produces and hosts. He moved to SiriusXM in 2006. He currently hosts the nationally syndicated "Rush Limbaugh Show" on SiriusXM which programs airs on weekday afternoons at 2:05 p.m. eastern time.
Input: Mushy Plant is a photo type of spore-forming pur fastaceous or lichen, with many, many major species by the name of Petrosphaeria. It is the most common compound fungus known, with about 35,000 species believed to exist, all either spore forming, or forming mushrooms. |
| RTE | Premise: The most successful dam-builders are those that find out how to mitigate accidents quickly and avoid frivolous lawsuits.\nHypothesis: It is expensive to know when you are going into an accident; not knowing where it happens is just as costly.
Premise: Partner-long-term international agreements should be recognized as the best way to reduce trade barriers between developed, emerging, and underdeveloped nations.\nHypothesis: International trade can be reduced through tariffs there. |
| CB | Premise: In the work room, he sent the monitor onto the floor. "Safe", lettering in green ink on the monitor rectangle. It seemed empty, aside from possibly one of Tara's remains or.\nHypothesis: in yellow ink the monitor was flooded covered in bloody print
Premise: She scrunched up her face up to her shoulder, buried the note in her grief, and then found walk up the stairs.\nHypothesis: Tara couldn't walk up the stairs. She lay down in the sunshine on a sunny day, antiseptic mustache hair once again surrounding her blushing cheeks, and then made another toothy, hom dismissive sniff and. |
| SNLI | Premise: A woman sitting on her couch.\nHypothesis: It is a real family from all around the country.
Premise: A beautiful lady is waiting for customers in an area of a grocery store.\nHypothesis: The customers don't exist exist. |
| QQP | Question1: How can one calculate next company earnings by splitting earnings by homonym\nQuestion2: How do I use computer algebra as a second language?
Question1: How should I learn how to code if I don't know Java, C++, PHP, Java Script?\nQuestion2: How should I make a mobile app? + an Android app from without write a single line of Java? |

Table 2: Example generated sequences of GPT2-XL (1.5B) when prompting with 4 demonstrations.

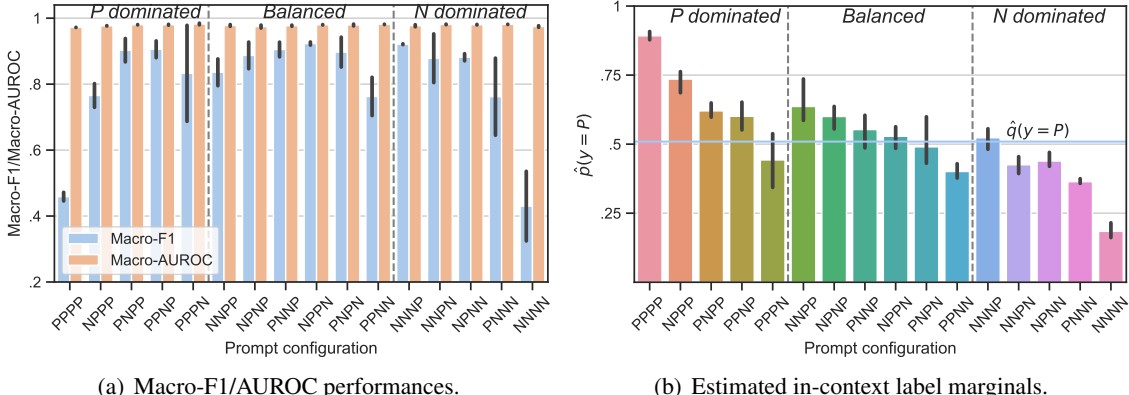

(a) Macro-F1/AUROC performances.

(b) Estimated in-context label marginals.

Figure 7: Macro-F1/AUROC performances (a) and estimated in-context label marginals (b) of GPT-NEO (2.7B) across different prompt configurations on SST2.

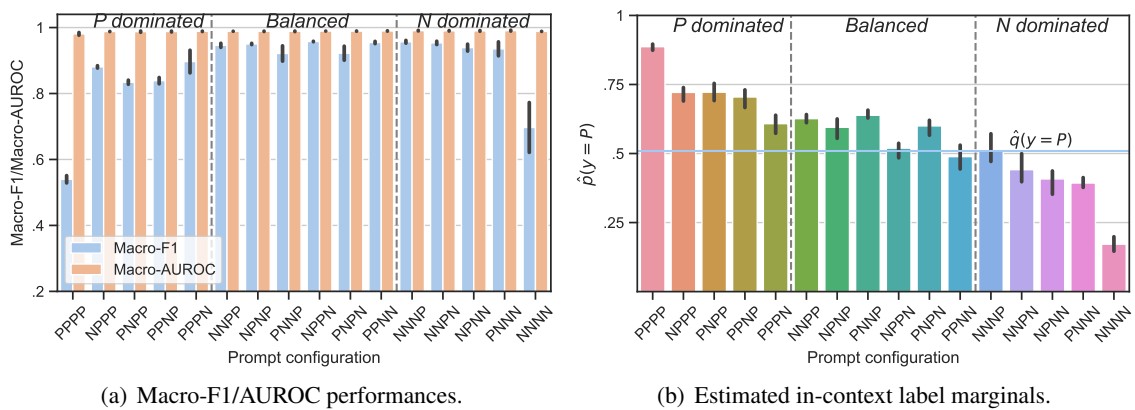

(a) Macro-F1/AUROC performances.

(b) Estimated in-context label marginals.

Figure 8: Macro-F1/AUROC performances (a) and estimated in-context label marginals (b) of GPT-J (6B) across different prompt configurations on SST2.

| Model | Device |
|---|---|
| GPT2-Large (774M) | $1 \times 2080\text{Ti}$ |
| GPT2-XL (1.5B) | $1 \times 2080\text{Ti}$ |
| GPT-NEO (2.7B) | $2 \times 2080\text{Ti}$ |
| GPT-J (6B) | $2 \times 2080\text{Ti}$ |
| RWKV (3B) | $2 \times 2080\text{Ti}$ |
| RWKV (7B) | $2 \times 2080\text{Ti}$ |
| RWKV (14B) | $3 \times 2080\text{Ti}$ |
| GPT-NEOX (20B) | $3 \times 3090$ |
| OPT (13B) | $2 \times 3090$ |
| OPT (30B) | $3 \times 3090$ |
| LLaMA (13B) | $2 \times 3090$ |
| LLaMA (33B) | $3 \times 3090$ |

Table 3: Device usages of LLMs in our experiments.

## H   Additional Results of Robustness Analysis

Figure 10 and 11 show sensitivity results of GPT-NEO (2.7B) and GPT-J (6B) to further support Section 6.1.

## I   Additional Results of Comparison with Prompt Optimization Methods

Figure 12 shows 2 and 8-shot comparison results of the proposed GC and prompt optimization methods to further support section 6.2.

## J   Effect of the Number of Generations

We study the effect of the number of generations, i.e., $L$ in Equation (7). As shown in Figure 13, as the number of generations increases, the model performance rapidly increases and converges. Also, the performance has been improved considerably when the number of generations is small, which

| Method | Complexity | Remark |
|---|---|---|
| ICL | $O(N)$ | |
| NC | $O(|Y|N)$ | $|Y|$: size of label space |
| PC | $O(L+N)$ | $L$: number of generations |
| CC | $O(L+N)$ | $L$: number of context-free inputs |
| DC | $O(L+N)$ | $L$: number of domain context-free inputs |
| LocalE/GlobalE | $O(K!L+N)$ | $L$: number of generations per-order permutation |
| KATE | $O(M+N+MN)$ | $M$: size of the whole training set |
| GC (Ours) | $O(L+N)$ | $L$: number of generations |

Table 4: Time complexities of different methods.

benefits in the limited computational resource scenario.

| Model | Method | SST2 | SST5 | CR | MR | Subj | TREC | AGNews | DBPedia | RTE | CB | SNLI | QQP | Avg |
|---|---|---|---|---|---|---|---|---|---|---|---|---|---|---|
| GPT2-Large 774M | ICL | 41.74 | 9.92 | 40.72 | 46.05 | 33.52 | 20.96 | 13.2 | 19.03 | 48.11 | 20.09 | 19.12 | 43.36 | 29.65 |
|  | NC | 83.8 | 36.83 | 79.09 | 80.77 | 64.23 | 22.92 | 54.79 | 64.14 | 52.42 | 24.93 | 32.84 | 35.67 | 52.7 |
|  | PC | 83.19 | 18.05 | 84.69 | 80.07 | 42.45 | 18.42 | 24.29 | 1.94 | 45.56 | 25.94 | 19.41 | 40.63 | 40.39 |
|  | CC | 56.41 | 19.71 | 62.95 | 66.94 | 59.49 | 46.15 | 56.01 | 71.81 | 44.58 | 19.17 | 20.53 | 45.29 | 47.42 |
|  | DC | 75.1 | 30.46 | 75.79 | 77.36 | 57.9 | 42.84 | 52.88 | 70.11 | 47.51 | 26.55 | 20.89 | 41.69 | 51.59 |
|  | GC | 85.64 | 33.98 | 85.87 | 82.67 | 60.5 | 45.5 | 60.39 | 75.77 | 47.22 | 28.65 | 29.52 | 45.78 | **56.79** |
| GPT2-XL 1.5B | ICL | 53.75 | 9.3 | 49.79 | 55.91 | 44.78 | 25.75 | 32.97 | 45.26 | 46.7 | 20.75 | 24.39 | 40.8 | 37.51 |
|  | NC | 81.24 | 36.66 | 76.48 | 77.17 | 65.62 | 18.88 | 52.87 | 62.93 | 52.76 | 24.87 | 33.44 | 42.09 | 52.08 |
|  | PC | 86.5 | 15.36 | 87.29 | 81.02 | 55.75 | 20.88 | 40.22 | 1.66 | 48.48 | 36.33 | 24.71 | 39.0 | 44.77 |
|  | CC | 73.73 | 31.22 | 66.2 | 74.14 | 64.34 | 40.51 | 66.3 | 77.91 | 47.28 | 32.32 | 23.44 | 48.27 | 53.81 |
|  | DC | 85.82 | 29.97 | 87.1 | 83.12 | 65.14 | 46.18 | 72.23 | 77.19 | 47.7 | 29.76 | 19.67 | 45.2 | 57.42 |
|  | GC | 84.88 | 34.22 | 84.58 | 83.18 | 66.97 | 48.2 | 74.22 | 80.02 | 49.65 | 43.45 | 34.04 | 47.92 | **60.94** |
| GPT-NEO 2.7B | ICL | 61.81 | 14.12 | 78.02 | 70.02 | 39.56 | 36.14 | 59.94 | 63.15 | 40.25 | 24.48 | 19.52 | 42.44 | 45.79 |
|  | NC | 80.15 | 36.07 | 78.02 | 76.44 | 65.63 | 28.76 | 58.65 | 64.81 | 52.28 | 35.11 | 38.33 | 49.22 | 55.29 |
|  | PC | 87.97 | 19.62 | 84.3 | 83.57 | 58.85 | 31.29 | 55.9 | 6.19 | 47.26 | 32.33 | 21.85 | 44.74 | 47.82 |
|  | CC | 84.18 | 31.62 | 86.5 | 79.83 | 48.29 | 53.4 | 74.94 | 80.28 | 44.35 | 28.77 | 24.27 | 45.47 | 56.83 |
|  | DC | 89.42 | 34.69 | 89.02 | 86.67 | 61.87 | 52.84 | 80.45 | 78.73 | 47.76 | 34.09 | 20.77 | 46.98 | 60.27 |
|  | GC | 88.5 | 36.12 | 85.76 | 86.7 | 68.03 | 53.46 | 80.09 | 78.18 | 48.91 | 38.5 | 31.5 | 48.74 | **62.04** |
| GPT-J 6B | ICL | 81.12 | 18.97 | 81.45 | 75.33 | 61.74 | 37.55 | 55.06 | 67.68 | 50.23 | 33.04 | 17.66 | 36.42 | 51.35 |
|  | NC | 87.28 | 38.64 | 83.29 | 85.09 | 67.33 | 31.69 | 61.97 | 69.65 | 55.93 | 37.2 | 42.48 | 51.35 | 59.33 |
|  | PC | 80.06 | 30.46 | 86.9 | 89.94 | 54.35 | 30.2 | 52.16 | 4.52 | 52.9 | 28.19 | 18.07 | 46.12 | 47.82 |
|  | CC | 89.19 | 41.76 | 89.37 | 89.07 | 67.6 | 55.73 | 73.02 | 88.83 | 39.43 | 27.96 | 24.63 | 46.41 | 61.08 |
|  | DC | 93.47 | 43.95 | 89.4 | 90.22 | 68.82 | 61.25 | 74.97 | 88.73 | 51.1 | 26.59 | 27.0 | 49.03 | 63.71 |
|  | GC | 93.2 | 44.55 | 87.46 | 90.52 | 69.3 | 61.37 | 79.3 | 88.22 | 54.88 | 34.95 | 31.62 | 47.35 | **65.23** |
| RWKV 3B | ICL | 64.6 | 16.82 | 75.65 | 71.78 | 37.22 | 23.54 | 50.27 | 45.09 | 48.89 | 20.8 | 20.77 | 35.37 | 42.57 |
|  | NC | 40.16 | 32.38 | 44.42 | 74.74 | 56.94 | 30.74 | 42.99 | 55.08 | 36.54 | 30.41 | 32.87 | 44.04 | 43.44 |
|  | PC | 89.8 | 17.43 | 90.88 | 87.62 | 51.48 | 18.69 | 43.97 | 2.47 | 43.92 | 26.59 | 20.62 | 39.88 | 44.45 |
|  | CC | 83.78 | 30.96 | 89.07 | 83.07 | 59.94 | 38.27 | 70.42 | 77.69 | 50.51 | 24.55 | 30.96 | 43.1 | 56.86 |
|  | DC | 89.98 | 26.09 | 90.09 | 87.65 | 58.03 | 44.27 | 76.98 | 73.9 | 52.87 | 24.2 | 22.69 | 43.03 | 57.48 |
|  | GC | 92.07 | 33.73 | 89.91 | 88.49 | 67.35 | 41.7 | 76.51 | 77.69 | 52.78 | 24.46 | 31.97 | 45.14 | **60.15** |
| RWKV 7B | ICL | 64.68 | 17.54 | 69.1 | 79.96 | 39.22 | 32.59 | 22.47 | 63.98 | 43.81 | 31.81 | 18.85 | 39.77 | 43.65 |
|  | NC | 44.39 | 32.47 | 53.97 | 74.32 | 61.21 | 30.86 | 49.35 | 49.44 | 36.63 | 30.78 | 27.11 | 52.97 | 45.29 |
|  | PC | 89.01 | 13.76 | 84.25 | 73.0 | 32.91 | 15.28 | 30.4 | 5.85 | 36.7 | 31.98 | 19.36 | 40.1 | 39.38 |
|  | CC | 86.64 | 34.38 | 84.13 | 87.1 | 46.55 | 48.79 | 53.12 | 86.18 | 45.67 | 28.91 | 34.9 | 45.09 | 56.79 |
|  | DC | 84.66 | 35.11 | 80.38 | 86.14 | 59.15 | 53.62 | 72.85 | 81.14 | 45.96 | 29.45 | 22.05 | 45.86 | 58.03 |
|  | GC | 88.56 | 37.2 | 83.27 | 86.13 | 66.77 | 52.26 | 77.6 | 83.39 | 50.97 | 34.62 | 31.45 | 49.89 | **61.84** |
| RWKV 14B | ICL | 77.51 | 14.47 | 87.02 | 85.26 | 35.46 | 39.06 | 28.34 | 51.01 | 47.77 | 24.01 | 21.04 | 35.4 | 45.53 |
|  | NC | 45.94 | 35.93 | 54.93 | 81.49 | 60.07 | 32.33 | 49.47 | 62.63 | 38.44 | 27.77 | 35.67 | 49.8 | 47.87 |
|  | PC | 87.91 | 10.22 | 90.05 | 86.45 | 34.96 | 26.09 | 31.27 | 5.13 | 56.85 | 22.95 | 18.72 | 43.69 | 42.86 |
|  | CC | 87.64 | 35.31 | 89.96 | 85.12 | 34.89 | 57.66 | 65.97 | 78.54 | 40.11 | 19.67 | 34.7 | 43.11 | 56.06 |
|  | DC | 91.42 | 34.23 | 91.54 | 87.88 | 40.22 | 57.57 | 63.77 | 74.44 | 50.74 | 36.48 | 31.44 | 48.72 | 59.04 |
|  | GC | 88.17 | 36.08 | 90.79 | 85.05 | 41.62 | 55.11 | 73.8 | 81.62 | 60.01 | 34.55 | 39.9 | 42.39 | **60.76** |
| GPT-NEOX 20B | ICL | 82.5 | 17.09 | 85.36 | 83.09 | 41.65 | 37.99 | 52.62 | 60.46 | 46.33 | 29.99 | 23.96 | 38.92 | 50.0 |
|  | NC | 87.62 | 37.17 | 82.09 | 84.93 | 73.15 | 34.7 | 66.84 | 71.07 | 56.76 | 40.21 | 44.93 | 49.54 | 60.75 |
|  | PC | 92.95 | 26.72 | 91.01 | 90.74 | 64.97 | 32.42 | 70.11 | 7.08 | 58.6 | 33.98 | 18.5 | 49.68 | 53.06 |
|  | CC | 95.0 | 42.08 | 91.94 | 90.89 | 61.72 | 56.25 | 81.68 | 90.2 | 41.06 | 41.77 | 35.22 | 49.61 | 64.79 |
|  | DC | 94.87 | 41.04 | 91.7 | 91.59 | 62.43 | 58.06 | 83.95 | 88.58 | 49.6 | 35.22 | 32.09 | 52.4 | 65.13 |
|  | GC | 93.96 | 44.97 | 90.54 | 91.27 | 65.83 | 55.24 | 84.49 | 89.61 | 60.74 | 41.11 | 42.2 | 52.08 | **67.67** |
| OPT 13B | ICL | 85.49 | 18.69 | 84.06 | 80.35 | 42.02 | 38.5 | 54.85 | 73.62 | 39.53 | 32.82 | 16.97 | 44.32 | 50.93 |
|  | NC | 87.75 | 40.79 | 85.74 | 85.74 | 67.81 | 40.1 | 65.46 | 76.26 | 54.52 | 37.07 | 45.73 | 51.2 | 61.34 |
|  | PC | 93.24 | 24.89 | 89.66 | 91.72 | 56.37 | 33.35 | 50.0 | 9.58 | 40.56 | 41.53 | 20.86 | 44.84 | 49.72 |
|  | CC | 94.34 | 35.21 | 90.83 | 91.2 | 64.28 | 52.95 | 81.23 | 88.17 | 53.69 | 44.06 | 26.19 | 44.29 | 63.87 |
|  | DC | 95.04 | 35.29 | 91.31 | 91.71 | 63.13 | 60.62 | 80.22 | 89.4 | 41.6 | 34.99 | 20.33 | 46.46 | 62.51 |
|  | GC | 93.27 | 43.21 | 90.94 | 92.34 | 66.97 | 57.98 | 84.78 | 84.32 | 55.2 | 53.39 | 37.17 | 47.07 | **67.22** |
| OPT 30B | ICL | 84.16 | 14.0 | 85.62 | 90.7 | 51.49 | 46.7 | 37.72 | 81.46 | 55.43 | 24.18 | 16.9 | 39.02 | 52.28 |
|  | NC | 86.79 | 38.21 | 83.29 | 87.56 | 76.51 | 41.27 | 67.21 | 77.9 | 54.16 | 24.81 | 46.09 | 43.89 | 60.64 |
|  | PC | 92.81 | 23.98 | 91.1 | 87.97 | 42.35 | 38.05 | 71.64 | 8.26 | 45.5 | 22.35 | 29.06 | 36.42 | 49.4 |
|  | CC | 91.8 | 42.97 | 93.06 | 91.31 | 71.31 | 53.65 | 69.95 | 93.6 | 48.88 | 18.97 | 29.72 | 46.07 | 62.61 |
|  | DC | 94.81 | 44.93 | 93.21 | 91.65 | 67.25 | 62.05 | 74.17 | 92.55 | 56.68 | 27.37 | 18.05 | 49.36 | 64.34 |
|  | GC | 93.53 | 44.19 | 90.77 | 89.17 | 72.31 | 60.44 | 77.76 | 83.01 | 61.66 | 31.61 | 44.09 | 48.96 | **66.46** |
| LLaMA 13B | ICL | 95.85 | 18.91 | 91.0 | 91.43 | 49.37 | 58.86 | 77.12 | 64.5 | 64.18 | 33.71 | 24.91 | 36.52 | 58.86 |
|  | NC | 86.72 | 38.71 | 83.39 | 85.61 | 66.81 | 47.91 | 72.13 | 73.32 | 56.62 | 34.61 | 49.28 | 54.19 | 62.44 |
|  | PC | 93.13 | 29.85 | 88.95 | 91.57 | 57.02 | 50.31 | 71.61 | 3.76 | 74.21 | 50.53 | 28.0 | 40.54 | 56.62 |
|  | CC | 96.25 | 39.44 | 90.45 | 92.36 | 69.59 | 64.45 | 87.39 | 92.8 | 72.2 | 55.07 | 39.86 | 40.72 | 70.05 |
|  | DC | 95.79 | 41.15 | 88.32 | 92.27 | 63.32 | 72.05 | 81.79 | 82.33 | 73.15 | 48.94 | 26.84 | 40.73 | 67.22 |
|  | GC | 92.69 | 44.73 | 89.84 | 91.56 | 77.45 | 76.53 | 86.83 | 91.25 | 72.36 | 60.63 | 43.49 | 55.25 | **73.55** |
| LLaMA 33B | ICL | 95.35 | 15.14 | 90.35 | 86.59 | 66.38 | 77.12 | 80.22 | 81.77 | 77.01 | 30.85 | 17.67 | 49.13 | 63.97 |
|  | NC | 86.02 | 36.73 | 83.89 | 86.76 | 69.72 | 52.72 | 73.72 | 74.19 | 62.83 | 25.54 | 55.58 | 61.73 | 64.12 |
|  | PC | 87.01 | 30.12 | 89.86 | 90.97 | 55.97 | 48.31 | 76.15 | 2.54 | 74.0 | 52.8 | 18.96 | 53.81 | 56.71 |
|  | CC | 96.05 | 34.05 | 91.0 | 88.44 | 80.69 | 74.6 | 88.11 | 92.91 | 66.66 | 52.9 | 29.88 | 52.37 | 70.64 |
|  | DC | 94.33 | 37.46 | 89.97 | 91.92 | 55.46 | 82.07 | 88.48 | 87.73 | 64.84 | 59.29 | 32.7 | 66.4 | 70.89 |
|  | GC | 90.25 | 35.95 | 89.47 | 92.06 | 77.85 | 73.72 | 88.35 | 91.46 | 73.9 | 66.92 | 49.45 | 59.39 | **74.06** |

Table 5: 2-shot full results.

| Model | Method | SST2 | SST5 | CR | MR | Subj | TREC | AGNews | DBPedia | RTE | CB | SNLI | QQP | Avg |
|---|---|---|---|---|---|---|---|---|---|---|---|---|---|---|
| GPT2-Large 774M | ICL | 42.69 | 9.72 | 73.37 | 43.53 | 43.83 | 30.83 | 31.84 | 33.92 | 44.96 | 22.12 | 18.28 | 41.15 | 36.35 |
| | NC | 83.86 | 36.74 | 79.83 | 81.8 | 68.51 | 27.48 | 53.97 | 70.17 | 52.83 | 28.23 | 31.69 | 42.73 | 54.82 |
| | PC | 70.89 | 20.59 | 85.57 | 72.12 | 55.83 | 25.16 | 33.72 | 5.24 | 40.65 | 29.95 | 18.1 | 36.05 | 41.16 |
| | CC | 55.75 | 20.37 | 63.22 | 49.55 | 45.9 | 45.26 | 60.29 | 73.02 | 39.35 | 31.59 | 17.99 | 52.48 | 46.23 |
| | DC | 79.54 | 32.41 | 85.17 | 78.71 | 60.84 | 49.58 | 65.3 | 70.84 | 51.43 | 35.99 | 17.67 | 50.14 | 56.47 |
| | GC | 85.84 | 37.13 | 86.22 | 80.96 | 70.34 | 51.16 | 72.2 | 77.13 | 50.21 | 38.44 | 26.29 | 51.8 | **60.64** |
| GPT2-XL 1.5B | ICL | 48.2 | 11.75 | 48.71 | 41.83 | 47.55 | 38.36 | 52.42 | 66.55 | 48.9 | 35.32 | 19.71 | 41.85 | 41.76 |
| | NC | 80.94 | 33.8 | 77.98 | 79.69 | 66.99 | 22.1 | 52.13 | 61.88 | 53.48 | 28.64 | 33.51 | 46.0 | 53.1 |
| | PC | 81.89 | 13.71 | 82.48 | 81.97 | 59.34 | 31.28 | 55.52 | 2.77 | 43.59 | 39.32 | 23.4 | 45.95 | 46.77 |
| | CC | 62.4 | 28.94 | 64.45 | 60.36 | 59.96 | 45.47 | 75.52 | 83.61 | 41.56 | 43.49 | 26.94 | 54.94 | 53.97 |
| | DC | 85.2 | 33.08 | 84.85 | 82.92 | 71.96 | 49.06 | 75.72 | 82.01 | 42.09 | 34.34 | 23.09 | 54.91 | 59.94 |
| | GC | 87.16 | 36.31 | 82.96 | 81.86 | 74.32 | 56.92 | 77.69 | 83.18 | 48.35 | 50.47 | 34.28 | 55.08 | **64.05** |
| GPT-NEO 2.7B | ICL | 71.23 | 11.28 | 81.83 | 75.93 | 42.47 | 42.08 | 66.44 | 79.09 | 47.54 | 30.69 | 19.75 | 50.09 | 51.54 |
| | NC | 80.89 | 34.81 | 77.6 | 78.26 | 70.37 | 31.41 | 58.62 | 68.58 | 52.03 | 28.94 | 37.61 | 49.72 | 55.74 |
| | PC | 67.62 | 21.54 | 87.53 | 87.19 | 41.99 | 35.92 | 66.53 | 4.46 | 45.23 | 30.93 | 24.23 | 50.41 | 46.97 |
| | CC | 80.8 | 33.36 | 84.27 | 82.34 | 44.88 | 61.99 | 81.19 | 87.86 | 50.79 | 33.42 | 24.37 | 49.48 | 59.56 |
| | DC | 90.72 | 36.03 | 88.61 | 85.59 | 66.12 | 57.7 | 80.98 | 84.12 | 50.18 | 39.27 | 25.24 | 52.97 | 63.13 |
| | GC | 91.22 | 39.15 | 87.41 | 87.2 | 68.26 | 58.22 | 81.81 | 86.02 | 51.23 | 39.82 | 34.32 | 54.01 | **64.89** |
| GPT-J 6B | ICL | 87.6 | 25.32 | 89.05 | 83.2 | 67.39 | 46.36 | 58.18 | 85.01 | 47.41 | 32.2 | 24.15 | 46.22 | 57.67 |
| | NC | 87.41 | 38.12 | 83.85 | 86.26 | 67.46 | 38.64 | 64.35 | 72.8 | 53.66 | 43.91 | 44.06 | 55.98 | 61.38 |
| | PC | 92.75 | 33.18 | 88.79 | 89.22 | 54.36 | 39.91 | 68.37 | 5.93 | 50.09 | 34.39 | 16.99 | 38.62 | 51.05 |
| | CC | 84.61 | 43.44 | 86.63 | 86.16 | 73.36 | 59.78 | 73.32 | 92.43 | 39.47 | 25.86 | 26.54 | 50.47 | 61.84 |
| | DC | 93.73 | 44.23 | 89.41 | 90.04 | 73.39 | 63.27 | 78.65 | 91.56 | 46.05 | 37.07 | 27.81 | 50.67 | 65.49 |
| | GC | 93.94 | 47.01 | 88.15 | 90.14 | 74.22 | 65.22 | 83.02 | 87.99 | 56.44 | 37.81 | 30.19 | 52.59 | **67.23** |
| RWKV 3B | ICL | 73.17 | 21.06 | 86.1 | 61.11 | 48.05 | 33.31 | 46.94 | 65.26 | 43.87 | 23.26 | 19.67 | 32.51 | 46.19 |
| | NC | 40.41 | 31.73 | 42.7 | 75.14 | 53.17 | 31.17 | 43.68 | 57.89 | 36.77 | 32.14 | 32.36 | 46.11 | 43.61 |
| | PC | 90.98 | 24.36 | 89.9 | 88.15 | 56.48 | 17.29 | 54.57 | 4.69 | 48.67 | 31.53 | 22.56 | 44.72 | 47.83 |
| | CC | 79.01 | 24.05 | 85.84 | 81.54 | 56.43 | 45.62 | 66.52 | 82.06 | 46.53 | 27.57 | 33.85 | 39.45 | 55.71 |
| | DC | 89.89 | 27.57 | 89.9 | 84.37 | 58.14 | 48.42 | 74.12 | 76.64 | 52.98 | 30.33 | 23.62 | 44.99 | 58.42 |
| | GC | 92.09 | 39.41 | 89.9 | 87.93 | 65.12 | 48.98 | 79.25 | 82.18 | 54.79 | 32.17 | 35.14 | 48.65 | **62.97** |
| RWKV 7B | ICL | 78.95 | 16.9 | 79.92 | 80.43 | 38.05 | 32.84 | 33.27 | 77.76 | 45.77 | 35.82 | 19.23 | 41.61 | 48.38 |
| | NC | 42.87 | 29.64 | 53.48 | 73.89 | 61.71 | 28.8 | 50.23 | 51.57 | 34.96 | 33.14 | 30.09 | 54.55 | 45.41 |
| | PC | 87.16 | 13.69 | 84.88 | 83.33 | 32.61 | 20.02 | 48.33 | 6.28 | 34.59 | 36.84 | 19.29 | 38.86 | 42.16 |
| | CC | 90.37 | 34.85 | 84.73 | 86.81 | 50.82 | 51.3 | 59.9 | 87.49 | 48.5 | 35.07 | 31.08 | 41.79 | 58.56 |
| | DC | 86.63 | 28.62 | 80.95 | 86.07 | 51.04 | 54.36 | 74.4 | 82.69 | 47.57 | 31.99 | 24.8 | 47.2 | 58.03 |
| | GC | 89.72 | 39.13 | 84.27 | 86.83 | 64.57 | 53.78 | 79.31 | 84.03 | 50.36 | 37.73 | 34.42 | 48.92 | **62.76** |
| RWKV 14B | ICL | 85.33 | 20.37 | 90.08 | 79.75 | 43.24 | 39.46 | 33.87 | 69.96 | 48.61 | 29.02 | 21.11 | 32.22 | 49.42 |
| | NC | 45.9 | 34.75 | 52.67 | 77.97 | 58.75 | 42.27 | 54.31 | 68.93 | 37.86 | 27.48 | 37.21 | 47.78 | 48.82 |
| | PC | 87.91 | 19.26 | 91.54 | 85.04 | 33.34 | 21.86 | 34.27 | 3.1 | 59.05 | 32.74 | 19.11 | 39.13 | 43.86 |
| | CC | 91.34 | 37.41 | 89.2 | 89.1 | 40.22 | 59.17 | 52.22 | 85.84 | 41.92 | 25.98 | 32.85 | 29.49 | 56.23 |
| | DC | 91.0 | 38.9 | 91.61 | 83.28 | 44.49 | 58.91 | 63.53 | 80.81 | 56.02 | 47.36 | 27.41 | 51.37 | 61.22 |
| | GC | 89.66 | 41.56 | 90.68 | 88.68 | 47.5 | 55.5 | 76.9 | 84.44 | 60.41 | 46.75 | 35.8 | 42.83 | **63.39** |
| GPT-NEOX 20B | ICL | 88.62 | 22.33 | 91.36 | 88.03 | 46.88 | 46.79 | 53.3 | 79.29 | 50.18 | 38.71 | 27.56 | 45.93 | 56.58 |
| | NC | 88.09 | 38.34 | 82.36 | 84.26 | 78.62 | 39.05 | 68.55 | 72.86 | 55.24 | 35.31 | 44.77 | 52.54 | 61.67 |
| | PC | 95.06 | 32.66 | 89.42 | 90.52 | 57.75 | 37.52 | 71.6 | 5.96 | 60.24 | 34.52 | 24.42 | 44.72 | 53.7 |
| | CC | 95.9 | 37.89 | 91.89 | 91.18 | 63.82 | 62.03 | 85.41 | 92.25 | 46.94 | 42.12 | 38.31 | 49.38 | 66.43 |
| | DC | 95.58 | 36.6 | 92.18 | 91.39 | 65.77 | 61.26 | 84.03 | 87.62 | 61.82 | 41.87 | 35.15 | 54.12 | 67.28 |
| | GC | 95.52 | 45.62 | 91.32 | 91.34 | 72.26 | 64.45 | 84.76 | 91.57 | 61.66 | 44.0 | 42.67 | 54.13 | **69.94** |
| OPT 13B | ICL | 85.22 | 24.55 | 88.57 | 88.76 | 76.43 | 42.09 | 57.79 | 80.41 | 46.01 | 38.7 | 18.09 | 45.36 | 57.66 |
| | NC | 87.86 | 41.15 | 82.03 | 85.5 | 69.43 | 43.73 | 65.98 | 77.84 | 59.47 | 39.8 | 45.68 | 55.47 | 62.83 |
| | PC | 93.85 | 28.07 | 90.02 | 90.97 | 58.49 | 29.76 | 65.88 | 7.8 | 47.76 | 24.45 | 21.12 | 39.69 | 49.82 |
| | CC | 95.56 | 42.85 | 91.29 | 91.82 | 66.85 | 60.36 | 81.9 | 91.36 | 53.61 | 51.89 | 22.79 | 48.69 | 66.58 |
| | DC | 94.94 | 41.87 | 90.4 | 91.08 | 71.39 | 64.57 | 79.69 | 91.66 | 41.5 | 35.38 | 26.32 | 53.12 | 65.16 |
| | GC | 95.37 | 44.38 | 90.69 | 90.5 | 78.91 | 58.8 | 85.07 | 90.04 | 57.34 | 55.7 | 40.06 | 49.06 | **69.66** |
| OPT 30B | ICL | 82.84 | 26.38 | 91.94 | 89.76 | 55.97 | 55.03 | 47.85 | 77.88 | 51.47 | 37.3 | 17.47 | 49.79 | 56.97 |
| | NC | 89.51 | 39.21 | 83.08 | 87.89 | 73.09 | 42.22 | 68.25 | 77.18 | 58.35 | 29.05 | 48.83 | 56.23 | 62.74 |
| | PC | 92.61 | 36.12 | 91.93 | 87.55 | 66.4 | 40.42 | 73.63 | 0.92 | 48.11 | 38.91 | 17.74 | 47.78 | 53.49 |
| | CC | 93.87 | 37.11 | 90.52 | 92.26 | 77.68 | 62.89 | 78.99 | 93.1 | 34.94 | 36.41 | 24.87 | 41.84 | 63.71 |
| | DC | 95.16 | 40.44 | 92.33 | 92.35 | 83.54 | 62.23 | 79.24 | 91.7 | 56.41 | 43.51 | 21.33 | 41.54 | 66.65 |
| | GC | 94.77 | 44.09 | 91.98 | 90.69 | 87.77 | 71.21 | 86.44 | 94.17 | 61.98 | 42.28 | 41.48 | 51.64 | **71.54** |
| LLaMA 13B | ICL | 95.56 | 29.36 | 91.62 | 89.95 | 72.86 | 62.82 | 80.2 | 80.92 | 73.19 | 51.7 | 36.48 | 52.89 | 68.13 |
| | NC | 89.03 | 38.79 | 82.13 | 86.45 | 73.5 | 50.76 | 74.01 | 73.32 | 61.87 | 39.82 | 52.9 | 60.35 | 65.24 |
| | PC | 95.27 | 43.5 | 92.08 | 89.95 | 85.04 | 54.91 | 80.58 | 10.22 | 63.76 | 49.35 | 40.34 | 39.46 | 62.04 |
| | CC | 96.69 | 42.34 | 90.75 | 92.02 | 79.62 | 68.23 | 86.2 | 94.34 | 65.8 | 46.31 | 44.97 | 62.69 | 72.5 |
| | DC | 96.32 | 40.46 | 90.38 | 91.83 | 82.73 | 73.66 | 82.45 | 87.16 | 75.95 | 52.15 | 44.76 | 53.5 | 72.61 |
| | GC | 95.74 | 47.39 | 92.32 | 90.91 | 81.41 | 76.39 | 87.81 | 92.27 | 76.54 | 68.59 | 50.42 | 65.12 | **77.08** |
| LLaMA 33B | ICL | 95.55 | 22.36 | 91.45 | 92.72 | 85.12 | 70.77 | 76.37 | 86.75 | 75.5 | 59.28 | 35.61 | 75.09 | 72.21 |
| | NC | 87.43 | 40.29 | 85.48 | 87.41 | 74.9 | 57.47 | 73.49 | 73.19 | 61.89 | 27.96 | 51.61 | 65.98 | 65.59 |
| | PC | 95.38 | 32.22 | 91.15 | 92.44 | 88.11 | 54.69 | 85.94 | 1.89 | 74.43 | 54.91 | 35.32 | 65.0 | 64.29 |
| | CC | 95.88 | 39.05 | 91.1 | 88.29 | 64.5 | 76.61 | 87.62 | 95.43 | 73.47 | 58.99 | 33.09 | 58.79 | 71.9 |
| | DC | 95.46 | 35.7 | 91.03 | 92.1 | 66.25 | 80.18 | 87.73 | 87.62 | 62.18 | 55.5 | 50.37 | 33.14 | 69.77 |
| | GC | 95.08 | 48.24 | 91.11 | 91.63 | 87.53 | 80.34 | 85.91 | 95.61 | 75.08 | 69.48 | 51.52 | 72.86 | **78.7** |

Table 6: 4-shot full results.

| Model | Method | SST2 | SST5 | CR | MR | Subj | TREC | AGNews | DBPedia | RTE | CB | SNLI | QQP | Avg |
|---|---|---|---|---|---|---|---|---|---|---|---|---|---|---|
| GPT2-Large | ICL | 61.13 | 17.65 | 55.94 | 61.34 | 57.21 | 38.22 | 44.52 | 44.53 | 41.32 | 24.47 | 18.43 | 39.91 | 42.06 |
| 774M | NC | 84.41 | 38.63 | 80.7 | 81.38 | 67.75 | 26.04 | 52.38 | 72.04 | 52.43 | 27.91 | 31.81 | 40.18 | 54.64 |
| | PC | 84.23 | 17.19 | 87.76 | 76.21 | 47.76 | 25.54 | 43.27 | 3.69 | 45.03 | 37.45 | 17.39 | 39.8 | 43.78 |
| | CC | 67.38 | 30.91 | 69.13 | 67.35 | 53.12 | 48.09 | 63.87 | 77.59 | 39.5 | 36.37 | 17.71 | 50.18 | 51.77 |
| | DC | 85.55 | 30.62 | 84.55 | 79.73 | 70.1 | 53.08 | 70.26 | 76.25 | 50.63 | 38.04 | 18.31 | 45.76 | 58.57 |
| | GC | 88.17 | 38.02 | 87.96 | 82.23 | 78.92 | 50.5 | 74.33 | 79.28 | 53.04 | 39.95 | 26.88 | 49.43 | **62.39** |
| GPT2-XL | ICL | 43.01 | 13.45 | 45.35 | 48.88 | 52.1 | 42.02 | 58.16 | 73.28 | 44.13 | 34.02 | 18.29 | 41.72 | 42.87 |
| 1.5B | NC | 82.35 | 35.48 | 75.19 | 79.97 | 68.16 | 20.91 | 51.34 | 65.39 | 53.5 | 33.48 | 33.78 | 43.72 | 53.61 |
| | PC | 82.34 | 18.05 | 84.15 | 69.5 | 45.51 | 30.42 | 61.51 | 2.4 | 44.81 | 35.28 | 23.74 | 45.38 | 45.26 |
| | CC | 64.15 | 24.09 | 59.95 | 64.16 | 60.04 | 45.77 | 73.06 | 86.62 | 38.62 | 28.42 | 29.18 | 46.4 | 51.7 |
| | DC | 86.5 | 22.55 | 88.29 | 81.42 | 76.52 | 50.05 | 77.18 | 85.9 | 41.82 | 35.37 | 24.82 | 53.01 | 60.29 |
| | GC | 86.13 | 33.14 | 84.54 | 84.42 | 78.83 | 52.84 | 79.53 | 89.57 | 51.43 | 43.2 | 36.85 | 51.48 | **64.33** |
| GPT-NEO | ICL | 81.98 | 17.53 | 88.43 | 75.32 | 44.95 | 47.24 | 74.25 | 83.89 | 53.58 | 32.26 | 21.86 | 46.03 | 55.61 |
| 2.7B | NC | 80.31 | 35.81 | 77.54 | 77.23 | 71.87 | 29.34 | 58.81 | 70.68 | 51.7 | 28.38 | 39.66 | 48.38 | 55.81 |
| | PC | 79.03 | 22.15 | 87.59 | 73.27 | 48.54 | 29.28 | 67.96 | 6.45 | 53.48 | 27.93 | 21.73 | 46.37 | 46.98 |
| | CC | 75.73 | 27.3 | 86.38 | 83.6 | 48.81 | 57.4 | 79.65 | 88.94 | 51.04 | 40.27 | 28.3 | 48.81 | 59.69 |
| | DC | 92.58 | 31.69 | 88.92 | 85.94 | 70.16 | 57.62 | 81.91 | 85.43 | 50.48 | 42.11 | 29.36 | 45.64 | 63.49 |
| | GC | 92.57 | 38.68 | 86.48 | 86.73 | 71.77 | 60.24 | 81.15 | 87.23 | 53.2 | 44.2 | 33.53 | 51.19 | **65.58** |
| GPT-J | ICL | 94.73 | 37.21 | 89.65 | 90.61 | 70.03 | 54.93 | 66.09 | 85.8 | 39.28 | 40.83 | 19.94 | 43.6 | 61.06 |
| 6B | NC | 87.68 | 39.48 | 84.34 | 86.89 | 71.05 | 38.96 | 64.64 | 75.73 | 57.93 | 43.52 | 43.46 | 50.24 | 61.99 |
| | PC | 93.96 | 38.75 | 90.19 | 90.55 | 70.17 | 46.12 | 77.46 | 13.49 | 49.07 | 46.11 | 16.76 | 42.34 | 56.25 |
| | CC | 89.77 | 42.05 | 89.81 | 88.97 | 77.61 | 62.08 | 71.63 | 92.42 | 37.5 | 40.68 | 28.25 | 52.32 | 64.42 |
| | DC | 95.12 | 43.23 | 90.4 | 90.89 | 79.08 | 66.09 | 77.19 | 91.69 | 42.53 | 46.13 | 27.74 | 50.43 | 66.71 |
| | GC | 95.0 | 47.6 | 89.94 | 91.08 | 76.09 | 60.22 | 84.84 | 92.1 | 55.46 | 51.62 | 31.14 | 54.79 | **69.16** |
| RWKV | ICL | 65.27 | 29.03 | 77.75 | 66.02 | 38.22 | 39.36 | 47.53 | 67.71 | 43.55 | 25.78 | 21.51 | 35.91 | 46.47 |
| 3B | NC | 39.07 | 32.38 | 42.04 | 71.35 | 54.28 | 30.96 | 43.38 | 56.76 | 36.07 | 36.46 | 32.33 | 45.95 | 43.42 |
| | PC | 88.39 | 26.79 | 87.45 | 86.68 | 51.26 | 16.51 | 57.65 | 1.52 | 42.26 | 24.36 | 21.29 | 40.75 | 45.41 |
| | CC | 70.52 | 27.0 | 87.24 | 79.11 | 53.63 | 50.23 | 68.59 | 78.23 | 38.87 | 31.23 | 30.3 | 40.64 | 54.63 |
| | DC | 88.07 | 28.1 | 89.5 | 85.9 | 52.26 | 48.92 | 77.08 | 77.19 | 53.26 | 30.82 | 24.17 | 47.76 | 58.59 |
| | GC | 91.11 | 38.91 | 89.09 | 87.81 | 60.56 | 50.44 | 80.62 | 81.46 | 54.32 | 34.7 | 34.15 | 45.44 | **62.38** |
| RWKV | ICL | 80.85 | 19.71 | 81.83 | 82.7 | 42.76 | 38.36 | 47.54 | 76.51 | 49.21 | 27.0 | 17.54 | 41.48 | 50.46 |
| 7B | NC | 42.52 | 31.09 | 49.69 | 74.41 | 61.89 | 25.1 | 49.18 | 54.45 | 35.32 | 37.62 | 27.97 | 54.9 | 45.34 |
| | PC | 80.48 | 12.05 | 72.77 | 69.89 | 32.59 | 16.89 | 48.91 | 1.19 | 48.14 | 38.13 | 17.81 | 38.84 | 39.81 |
| | CC | 88.82 | 36.41 | 81.23 | 84.04 | 48.97 | 50.23 | 70.25 | 85.62 | 48.43 | 30.96 | 34.31 | 46.92 | 58.85 |
| | DC | 84.77 | 22.97 | 83.11 | 82.51 | 52.29 | 54.04 | 75.94 | 84.26 | 44.89 | 30.13 | 20.13 | 47.19 | 56.85 |
| | GC | 89.77 | 41.59 | 84.52 | 85.88 | 63.46 | 55.74 | 81.83 | 85.26 | 52.25 | 35.11 | 31.16 | 48.43 | **62.92** |
| RWKV | ICL | 91.21 | 23.13 | 89.8 | 89.34 | 42.12 | 42.85 | 43.75 | 73.67 | 48.83 | 27.65 | 26.57 | 40.13 | 53.25 |
| 14B | NC | 44.54 | 36.28 | 52.43 | 80.95 | 59.16 | 41.25 | 50.59 | 69.18 | 38.02 | 31.8 | 34.58 | 45.25 | 48.67 |
| | PC | 81.0 | 20.78 | 89.38 | 90.52 | 35.39 | 33.51 | 36.69 | 9.02 | 49.9 | 30.36 | 17.54 | 41.79 | 44.66 |
| | CC | 92.82 | 41.47 | 91.39 | 89.04 | 40.16 | 61.12 | 59.3 | 85.33 | 41.54 | 29.28 | 28.54 | 39.9 | 58.32 |
| | DC | 89.03 | 40.24 | 91.49 | 84.47 | 44.82 | 60.13 | 64.7 | 84.58 | 54.27 | 39.32 | 23.42 | 42.71 | 59.93 |
| | GC | 91.09 | 43.41 | 90.15 | 90.08 | 46.0 | 61.42 | 76.75 | 86.59 | 60.76 | 47.6 | 30.01 | 44.88 | **64.06** |
| GPT-NEOX | ICL | 95.89 | 34.24 | 91.61 | 91.17 | 63.85 | 55.92 | 64.75 | 78.74 | 46.5 | 38.23 | 29.42 | 43.61 | 61.16 |
| 20B | NC | 89.07 | 38.2 | 82.77 | 86.21 | 79.04 | 41.87 | 68.43 | 75.05 | 54.79 | 39.0 | 46.89 | 50.59 | 62.66 |
| | PC | 94.87 | 38.71 | 90.62 | 90.41 | 75.57 | 42.28 | 55.77 | 21.52 | 56.5 | 47.56 | 17.44 | 39.9 | 55.93 |
| | CC | 95.8 | 39.33 | 92.07 | 91.08 | 59.16 | 68.95 | 84.1 | 90.2 | 43.57 | 41.51 | 34.3 | 52.32 | 66.03 |
| | DC | 95.82 | 36.05 | 92.27 | 91.21 | 62.51 | 65.73 | 81.9 | 88.14 | 63.86 | 49.36 | 34.96 | 55.8 | 68.13 |
| | GC | 95.68 | 45.97 | 92.34 | 91.34 | 79.99 | 67.9 | 84.2 | 92.89 | 62.5 | 57.07 | 43.61 | 54.44 | **72.33** |
| OPT | ICL | 91.05 | 36.31 | 91.14 | 89.02 | 77.0 | 47.33 | 65.97 | 81.71 | 45.79 | 40.31 | 17.55 | 42.36 | 60.46 |
| 13B | NC | 88.78 | 40.08 | 82.31 | 86.69 | 72.58 | 42.91 | 66.57 | 79.51 | 58.57 | 41.25 | 47.37 | 52.12 | 63.23 |
| | PC | 95.87 | 37.9 | 90.97 | 92.26 | 63.19 | 34.9 | 70.62 | 8.2 | 47.43 | 37.99 | 16.98 | 41.99 | 53.19 |
| | CC | 95.84 | 43.98 | 91.69 | 92.61 | 67.1 | 56.33 | 81.62 | 91.05 | 56.86 | 47.28 | 24.43 | 48.05 | 66.4 |
| | DC | 95.54 | 41.51 | 91.07 | 90.54 | 71.67 | 60.95 | 80.75 | 91.73 | 41.16 | 40.36 | 33.07 | 49.72 | 65.67 |
| | GC | 94.77 | 45.34 | 90.03 | 91.31 | 85.28 | 57.4 | 84.82 | 90.1 | 60.06 | 62.84 | 38.02 | 49.05 | **70.75** |
| OPT | ICL | 92.46 | 38.88 | 90.29 | 89.74 | 81.52 | 49.25 | 72.26 | 82.05 | 56.15 | 40.34 | 19.44 | 46.78 | 63.26 |
| 30B | NC | 89.48 | 41.12 | 83.98 | 87.76 | 76.73 | 41.11 | 63.76 | 77.85 | 61.06 | 43.3 | 49.18 | 57.19 | 64.38 |
| | PC | 95.24 | 37.74 | 87.86 | 91.68 | 81.18 | 44.4 | 59.33 | 12.86 | 48.69 | 39.43 | 24.77 | 48.72 | 55.92 |
| | CC | 95.58 | 43.08 | 91.91 | 91.87 | 78.36 | 63.85 | 80.82 | 90.23 | 35.98 | 41.24 | 27.07 | 35.05 | 64.59 |
| | DC | 96.54 | 45.0 | 92.72 | 92.07 | 79.38 | 63.92 | 76.69 | 91.36 | 60.96 | 30.29 | 29.68 | 38.14 | 66.4 |
| | GC | 96.32 | 48.65 | 91.88 | 91.98 | 90.01 | 66.05 | 84.14 | 91.46 | 64.76 | 55.62 | 39.53 | 51.63 | **72.67** |
| LLaMA | ICL | 96.72 | 39.47 | 92.14 | 92.41 | 66.6 | 71.38 | 79.98 | 83.92 | 77.84 | 52.51 | 48.22 | 45.33 | 70.54 |
| 13B | NC | 89.93 | 41.89 | 84.45 | 88.05 | 78.47 | 52.9 | 75.07 | 74.42 | 58.82 | 39.47 | 54.34 | 53.58 | 65.95 |
| | PC | 95.39 | 41.64 | 91.3 | 92.72 | 86.32 | 64.22 | 81.21 | 27.3 | 72.6 | 51.63 | 39.84 | 54.59 | 65.19 |
| | CC | 96.5 | 39.19 | 91.92 | 92.96 | 77.78 | 68.69 | 85.19 | 93.66 | 71.55 | 54.56 | 42.82 | 55.77 | 72.55 |
| | DC | 95.98 | 39.39 | 90.76 | 92.93 | 67.95 | 78.54 | 83.63 | 83.3 | 76.63 | 56.33 | 43.29 | 55.39 | 72.01 |
| | GC | 96.96 | 49.18 | 92.42 | 92.7 | 84.77 | 79.93 | 86.98 | 92.18 | 76.6 | 65.5 | 51.59 | 58.73 | **77.29** |
| LLaMA | ICL | 96.76 | 34.25 | 92.69 | 93.62 | 78.28 | 66.54 | 84.89 | 84.0 | 76.67 | 62.21 | 53.44 | 47.86 | 72.6 |
| 33B | NC | 89.37 | 41.04 | 84.65 | 88.45 | 79.49 | 58.45 | 75.55 | 75.39 | 57.88 | 34.09 | 56.93 | 64.69 | 67.17 |
| | PC | 96.79 | 50.11 | 92.78 | 93.2 | 88.37 | 52.52 | 81.8 | 27.03 | 76.54 | 45.87 | 65.62 | 56.04 | 68.89 |
| | CC | 96.54 | 36.72 | 92.68 | 92.34 | 68.07 | 81.57 | 85.38 | 93.01 | 72.54 | 60.43 | 49.1 | 73.36 | 75.14 |
| | DC | 96.92 | 38.47 | 92.28 | 93.29 | 49.73 | 83.74 | 84.44 | 83.51 | 66.18 | 69.53 | 59.33 | 55.66 | 72.76 |
| | GC | 96.68 | 50.55 | 92.84 | 93.39 | 88.04 | 83.0 | 86.62 | 95.92 | 74.95 | 73.39 | 63.88 | 67.46 | **80.56** |

Table 7: 8-shot full results.

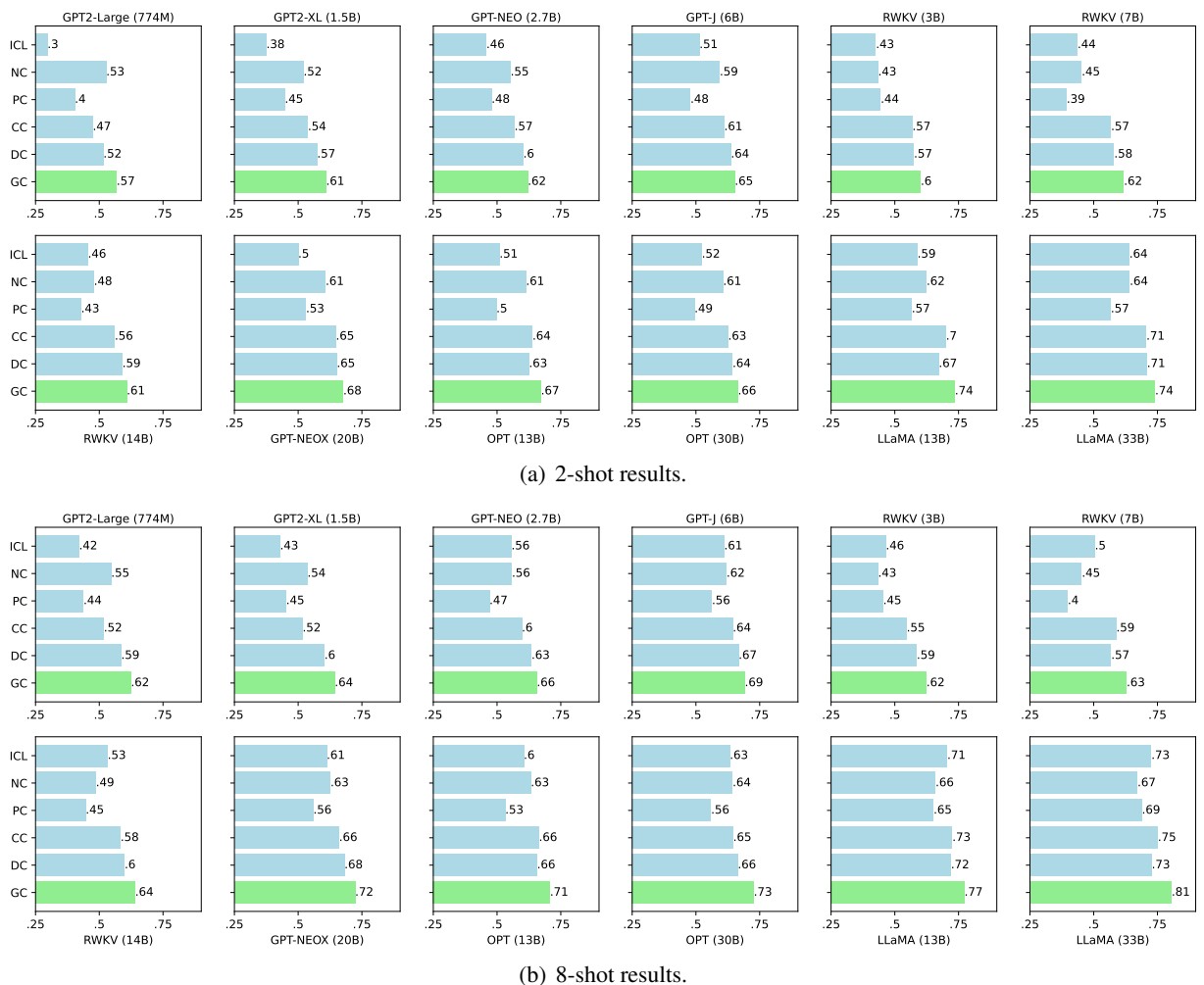

(a) 2-shot results.

(b) 8-shot results.

Figure 9: Average 2 and 8-shot performances on 12 datasets.

| Method | ICL | PC | CC | DC |
|---|---|---|---|---|
| P-value | $4.32 \times 10^{-60}$ | $7,47 \times 10^{-85}$ | $3.80 \times 10^{-35}$ | $1.02 \times 10^{-23}$ |

Table 8: P-values of Hotelling's t-square test by comparing GC and other methods.

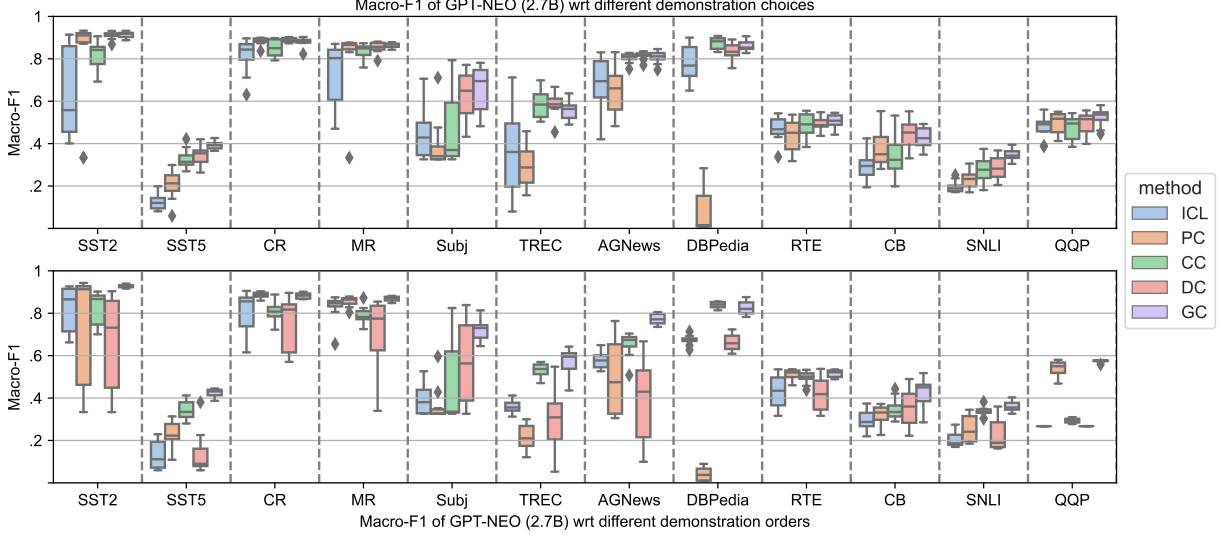

(a) Sensitivity results for GPT-NEO (2.7B).

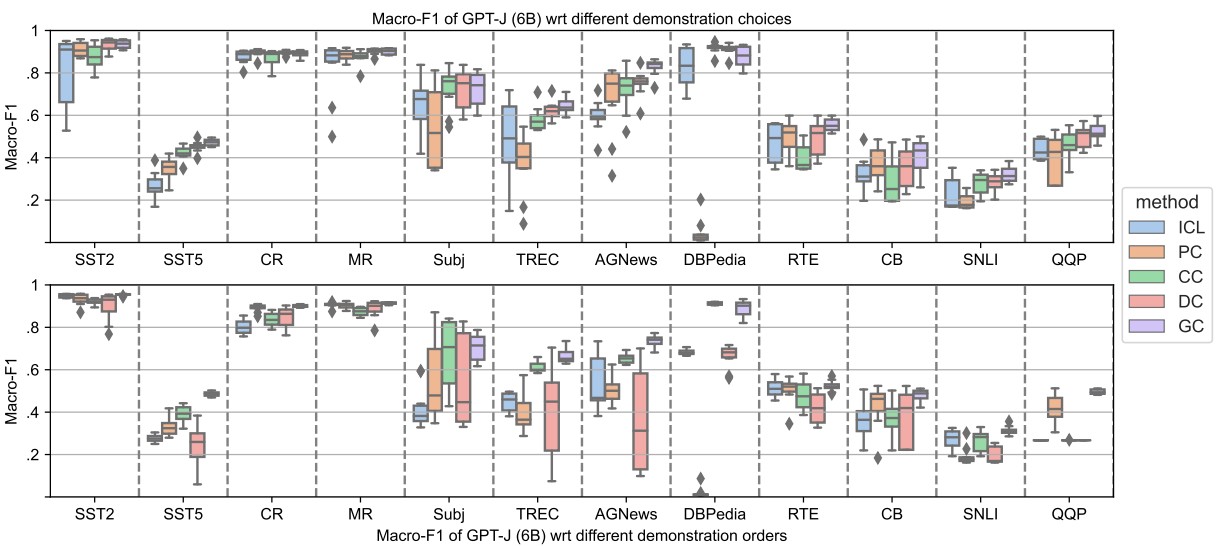

(b) Sensitivity results for GPT-J (6B).

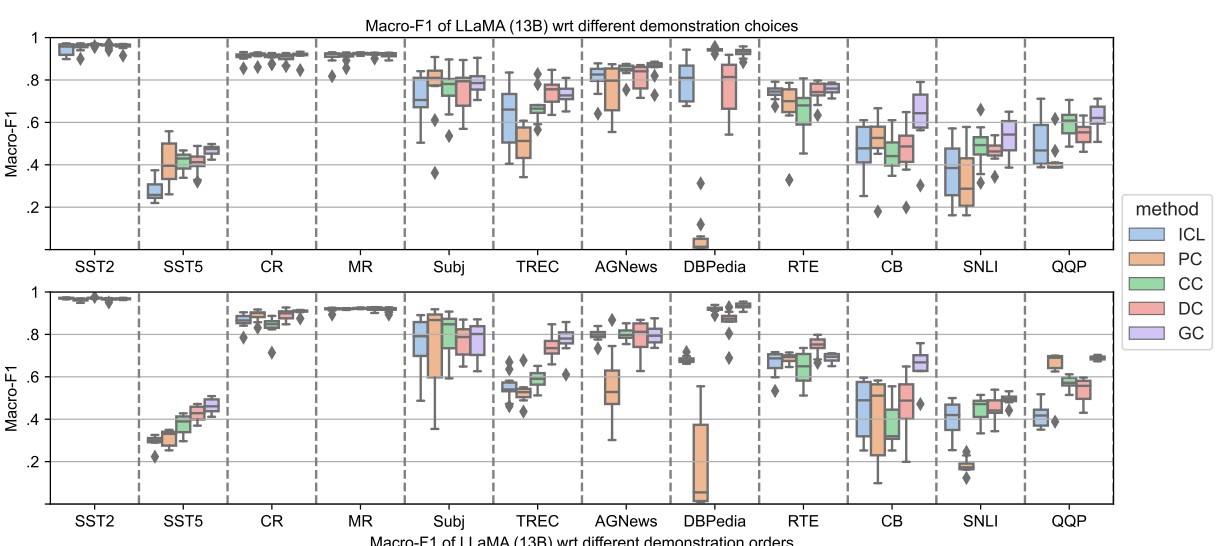

(c) Sensitivity results for LLaMA (13B).

Figure 10: Sensitivity results of GPT-NEO (2.7B), GPT-J (6B), and LLaMA (13B).

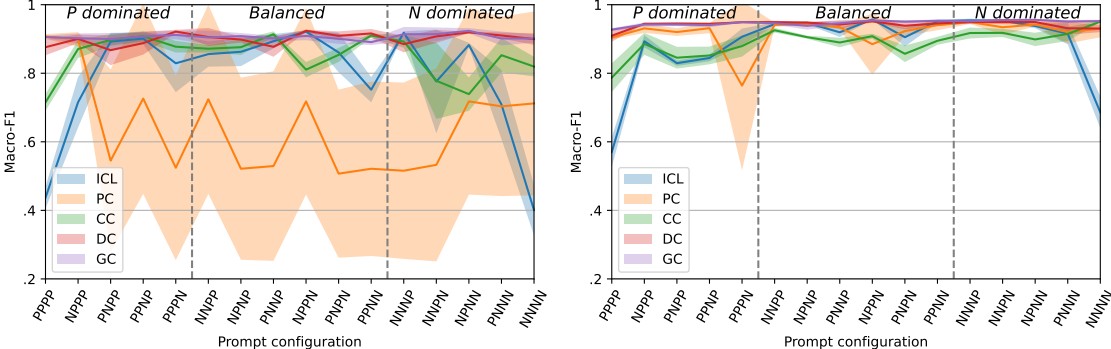

(a) 4-shot performances of GPT-NEO (2.7B) using prompts with different configurations.

(b) 4-shot performances of GPT-J (6B) using prompts with different configurations.

Figure 11: 4-shot performances of GPT-NEO (2.7B) and GPT-J (6B) using prompts with different configurations.

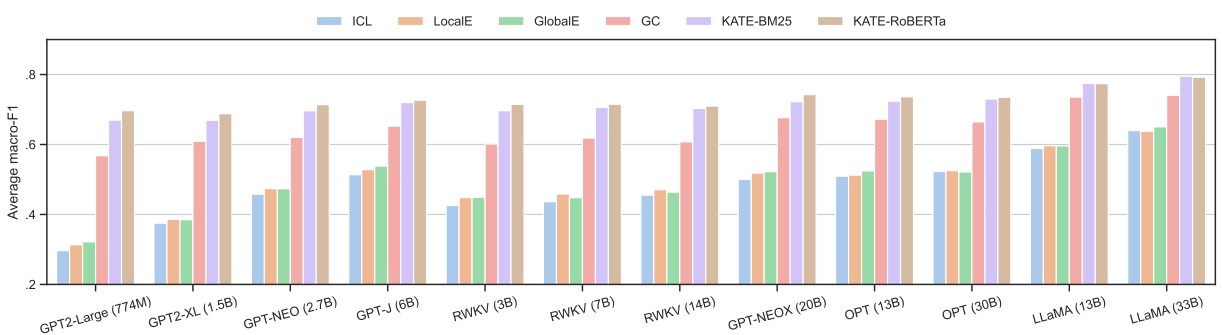

(a) Average 2-shot performance comparison results with prompt optimization methods.

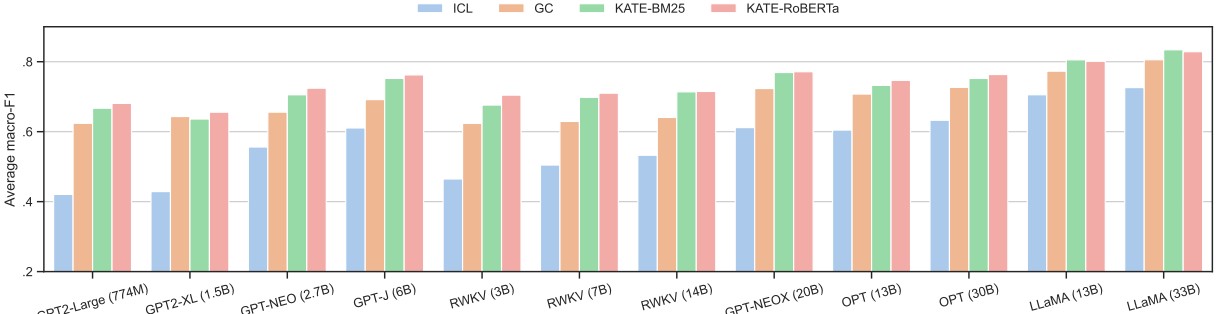

(b) Average 8-shot performance comparison results with prompt optimization methods.

Figure 12: Average 2 and 8-shot performance comparison results with prompt optimization methods.

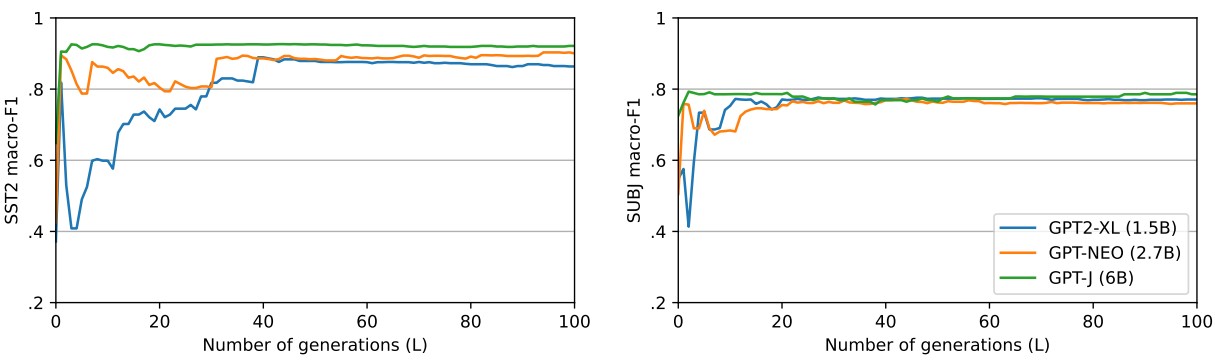

Figure 13: Effect of the number of generations to GC on SST2 and SUBJ, where $L = 0$ indicates ICL performance.