# OpenReview forum: "Generative Calibration for In-context Learning"
_EMNLP/2023/Conference — EMNLP 2023 Findings_

### Official Review · Reviewer_eRKA · 2023-07-23

**Soundness:** 3

**Excitement:**

4: Strong: This paper deepens the understanding of some phenomenon or lowers the barriers to an existing research direction.

**Paper Topic And Main Contributions:**

Authors propose a method of calibrating in-context learning (ICL) classifications by estimating generation probabilities. Authors empirically show that $p(x|y)$ is effective at ranking examples in terms of AUROC, and thus its only limitation is the lack of calibration. Authors propose to estimate the marginal probability $p(y)$ by generating samples and using Monte-Carlo estimation. Authors compare against a good range of existing methods for calibration, and demonstrate that the proposed method outperforms prior calibration methods consistently across various settings.

**Questions For The Authors:**

In equation (4), the second equation should be $\frac{p(N) p(P \mid x)}{p(P) p(N \mid x)}$?

On certain datasets, such as AGNews and SNLI, I believe the original dataset contains sufficient data to train $q(y \mid x)$ (say, finetune a BERT), $q(y)$ (simple statistical estimate would work), $q(x \mid y)$ (finetune LM on the data). Authors could use these to establish how well $p(x \mid y)$ estimates $q(x \mid y)$.

If we believe $p(x \mid y)$ is a good estimate, why can't we use $p(x \mid y) q(y) \approx p(x \mid y)$ for classification?

**Reasons To Accept:**

The idea is simple and straightforward, therefore it is likely to be a robust finding, and also provides good opportunities for follow-up studies to build upon.

Nonetheless, this simple idea (that in-context learning suffers from domain shift; i.e. marginals are poorly estimated, while conditionals are well-estimated) does not seem to have been clearly stated and empirically tested. This strengthens our understanding of in-context learning.

Experiments are fairly extensive; authors experiment with 12 text classification tasks, 12 LLMs, and three different few-shot settings. The empirical benefit of the proposed method seems to be quite consistent across all these settings.

**Reasons To Reject:**

Some of the statements lack scientific rigor.

In line 171, authors claim that according to Schwartz's theorem, posterior estimation would be poor when the number of samples K is small; but I believe the theorem is only a sufficient condition for the posterior to be good, and doesn't preclude small-sample estimation to be poor?

In line 174-193 authors argue that the estimation of $p(x, y)$ is hard because $p(\theta \mid \mathcal{D}_t)$ is hard to estimate (which is an assumption rather than a fact), but then suddenly jump to a conclusion that it will lead to label shift $p(y) \neq q(y)$. I don't see the logic connecting the assumption to the conclusion.

$p(x \mid y)$ showing high AUROC is not a sufficient condition for it to be close to $q(x \mid y)$. It can still be terribly ill-calibrated. There are potentially other ways of measuring the goodness of estimation (see Questions for Authors). At the least, authors should clearly state the limitation.

**Reproducibility:**

4: Could mostly reproduce the results, but there may be some variation because of sample variance or minor variations in their interpretation of the protocol or method.

**Reviewer Confidence:**

4: Quite sure. I tried to check the important points carefully. It's unlikely, though conceivable, that I missed something that should affect my ratings.

---

> ### Author Rebuttal · Authors · 2023-08-29
>
> We sincerely thank Reviewer eRKA for the valuable feedback. Now we will reply one by one according to the points mentioned by the reviewers.
>
> **Reasons to Reject#1:** Non-rigor description of Schwartz's theorem.
> - **We don't deny the reviewer's view on the Schwartz's theorem, but we also don't claim that posterior estimation would be poor when the number of samples K is small, which Reviewer eRKA may have some misunderstandings on this point.** In line 171, we just say that LLMs can't accept much training examples due to the limited context length, so according to Schwartz's theorem we can't ENSURE that the posterior is good. But we don't say that posterior would or must be poor when K is small, where neither the literal nor the extended meaning of this part can be understood in this way. So if there are some writing issues causing the misunderstanding, we apologize for this and we would modify the description to make it more clear and easier to follow.
>
> **Reasons to Reject#2:**
> - First of all, we need to clarify that the goal of in-context learning is estimate the true data distribution $q(x,y)$ (which is agnostic to us and we only have a few samples from it, i.e., the training dataset) using in-context generative distribution $p(x,y)$. (which is already in hand by prompting LLMs with a few training examples). So what do you mean to estimate $p(x,y)$? We feel like it is a clerical error of Reviewer eRKA.
>
>   **We believe Reviewer eRKA doesn't follow or understand line 174-193. Basically, this part is to show how the few-shot learning setting could affect the posterior $p(\theta|D_t)$, and then further affect the quality of $p(x,y|D_t)$ according to the Bayesian interpretation**. In specific, we propose two reasons of label-shift when the number of training examples $K$ is small, namely shifted prior and bad samples.
>
>   - Shifted prior (described in line 175-182): When $K$ is small, the prior $p(\theta)$ (the prior actually could be viewed as the topic's occurrence frequency in the pretrained corpus, since the latter is its MLE estimation) would dominate the posterior. This is because according to Equation (3), few product terms of $p(x_i,y_i|\theta)$ makes $p(\theta)$ dominate the posterior, and when there are no product terms (zero-shot) the posterior equals to the prior. **In this case, once the prior has a bias on a specific label token, it would cause a shift on this label.** For example, we have tested that GPT2-XL predicts positive on 74.32% of SST2 evaluation examples in the zero-shot setting, suggesting it has a label-shift on prior. This is reasonable because the token "positive" seems to appear more that token "negative" in the pretrained corpus. Other evidences include common token bias [1] and prediction insensitivity [2] when the training labels are corrupted studied in the previous works.
>
>   - Bad samples (described in line 182-193): To see the second reason, let's consider an example of review sentiment classification (labels are positive (P) and negative (N)). Since pretrained corpus contains diverse data from different websites, we can imagine that there not only exists the true topic $\theta^\star$ (for example, IMDB reviews page) such that the corresponding label marginal $p(y|\theta^\star)$ is uniform, i.e., $p(y=P|\theta^\star)=p(y=N|\theta^\star)=0.5$, but also exists a completely positive label-imbalanced topic $\theta'$ in the corpus (for example, IMDB reviews page filtered by ratings 5) such that $p(y=P|\theta')=1$ and $p(y=N|\theta')=0$. For any positive sample $x,P$, clearly $p(x,P|\theta')>0$ and $p(x,P|\theta^\star)>0$, since both topics support positive reviews. **Once our few-shot training dataset happens to ONLY contains positive samples, the posterior mass on $\theta'$, i.e., $p(\theta'|D_t)$ can be non-zero or even large, which would amplify the marginal distribution of the positive label and cause a label shift.** This is also called majority bias [1] in previous studies.
>
>   **Therefore, line 174-193 is actually sound and coherent in logic and not "suddenly jump to the conclusion"**. We think the reason why reviewer eRKA has confusion on this part is that we omit some details and leave deep understanding in the cited works. We apologize for this and will make it more clear in the next version.
>
> **Reasons to Reject#3**: $p(x|y)$ showing high AUROC is not a sufficient condition for it to be close to $q(x|y)$.
>
> - We admit that high AUROC is a necessary but not sufficient condition to take $p(x|y)$ be close to $q(x|y)$. Actually, we don't say so and we've used very careful wordings in the paper: we say in-context label conditional is good but NOT close to $q(x|y)$, where good means the model performs well in inter-label ranking, i.e., has a high AUROC.
>
>   **To the matter of fact, for our method to be effective, we don't need the model label conditional $p(x|y)$ to be close to the true one, we only need the model to perform well in inter-label ranking:** it can put most of the positive examples in the front, and most of the negative samples in the back (in a binary classification setting). If so, all we need is to set a proper decision threshold (an 1D decision boundary), which is actually what our calibration method do: if the model has shift on the positive label ($p(P)>p(N)$), our proposed generative calibration actually increases the decision threshold from the default 1 to $\frac{p(P)}{p(N)}$, i.e., predict $P$ when $\frac{p(P|x)}{p(N|x)}>\frac{p(P)}{p(N)}$ and predict $N$ otherwise. The merit of choosing this decision boundary is to ensure that the model label marginal is uniform but not biased to any specific labels.
>
>   Therefore, we use AUROC to measure the model's inter-label ranking performance to illustrate that $p(x|y)$ is good, which is enough for our proposed generative calibration to work. We apologize for the misleading description in the paper. Especially, we don't carefully explain what the word "good" means. We will revise this point in the next version.
>
> **Question#1:** Equation (4) is wrong?
>
> - No, Equation (4) is right. We can see it by simply using Bayesian formula:
> $$
> \frac{p(x|P)}{p(x|N)}=\frac{p(P)p(x|P)/p(x)}{p(N)p(x|N)/p(x)}=\frac{p(P)p(x|P)}{p(N)p(x|N)}
> $$
>
> **Question#2:** How well $p(x|y)$ estimates $q(x|y)$?
>
> - As discussed in the reply to Reasons to Reject#3, we don't actually need the in-context label conditional $p(x|y)$ to be close to the true one, so in the paper we don't estimate an oracle $q(x|y)$ and compare two distributions.
>
> **Question#3:** Why don't use $p(x|y)q(y)$?
> - We actually have mentioned this point in line 332-340. The problem is that we don't know the label marginal distribution $q(y)$ of the true scenario, i.e., the testing data. therefore we follow previous works to take $q(y)$ to be uniform. This predictor is equivalent to $p(x|y)=p(y|x)/p(y)$, where $p(y|x)$ is the original in-context predictor and $p(y)$ is the estimated marginal (via generation).
>
> **Clarification**
> - We find that much of concerns of Reviewer eRKA come from misunderstandings, some of which might cased by our writining issues. **We will revise the paper to make it more clear. We also sincerely hope that Reviwer eRKA can revisit to related parts and increase the soundness score, especially given the richness of our experiments and the effectiveness of our method (outperforms SOTA method DC in large margins) in different settings.**
>
> **References**
> 1. Zhao Z, Wallace E, Feng S, et al. Calibrate before use: Improving few-shot performance of language models[C]//International Conference on Machine Learning. PMLR, 2021: 12697-12706.
> 2. Min S, Lyu X, Holtzman A, et al. Rethinking the role of demonstrations: What makes in-context learning work?[J]. arXiv preprint arXiv:2202.12837, 2022.

---

### Official Review · Reviewer_Vz7F · 2023-07-31

**Typos Grammar Style And Presentation Improvements:** In Eq(4), p(P) should be the denomina…
**Soundness:** 3

**Excitement:**

4: Strong: This paper deepens the understanding of some phenomenon or lowers the barriers to an existing research direction.

**Paper Topic And Main Contributions:**


This paper proposes a label-shift calibration method for in-context learning after analysing the distribution shift theoretically and empirically. Especially, it first decomposes the posterior distribution of the generated sequence based on the assumption of the Bayesian process. Then, it derives the ratio of label condition distribution (positive/negative) as the odds of predicting as positive and further uses the AUROC to empirically show that the label distribution is almost unbiased. Next, it adopts the Monte-Carlo sampling methods to derive the label marginal distribution and shows the discrepancy between generated labels and that in the validation samples. Based on the observations, it proposes to use the multiple generation method to approximate the true label marginal distribution as the true posterior distribution is approximated.

Contributions:
It analysis the posterior distribution by decomposing it into label conditional distribution and label marginal distribution, which can be empirically calculated and used to verify the motivation.
The proposed method demonstrates improvement over the listed baselines under the paper settings.


**Questions For The Authors:**

1. More details about the implementation of “multiple generation”.
2. What is the time cost compared to ICL, with different sequence lengths and the number of generations (as shown in Figure 12, ICL is L=0)


**Reasons To Accept:**

1. The paper is well-written and the Bayesian-based analysis is clear and well-supported by the empirical results.
2. The research topic is important to the area and experimental results are inspiring as existing models are quite sensitive to the selected samples and their orders.


**Reasons To Reject:**

1. The proposed calibration method is not well explained. It would be better if more details are added about the “multiple-generation” solution.
2. Concerns about the efficiency of the solution as the model need to generate multiple times (if I understand it correctly), especially when the L(sequence length) is larger as its complexity is to be O(L), as well as a large number of generation.


**Reproducibility:**

2: Would be hard pressed to reproduce the results. The contribution depends on data that are simply not available outside the author's institution or consortium; not enough details are provided.

**Reviewer Confidence:**

4: Quite sure. I tried to check the important points carefully. It's unlikely, though conceivable, that I missed something that should affect my ratings.

---

> ### Author Rebuttal · Authors · 2023-08-29
>
> We sincerely thank Reviewer Vz7F for the valuable feedback. Now we will reply one by one according to the points mentioned by the reviewers.
>
> **Reasons to Reject#1:** The proposed calibration method is not well explained.
>
> - All we have to do compared to regular ICL is to estimate an additional label marginal distribution $p(y)$, which is nartually estimated by the Monte-Carlo sampling:
> $$
> p(y)=\sum_{x}p(x)p(y|x)\simeq\frac{1}{L}\sum_{i=1}^lp(y|x^l)
> $$
> where $l$ is the $l$-th sample from $p(x)$ and $L$ is the number of samples. Note that according to Equation (2), $p(x)$ is the continuation distribution given a prompt containing a few training examples. Therefore, the sample of $p(x)$ is naturally obtained by generating (sampling) sentences conditioned on that prompt.
>
>   **We sincerely thank for Reviewer Vz7F for pointing out that it is better to elaborate our method more detailedly, like presenting a figure, but we don't consider this issue to be strong enough as a reason to reject, because:**
>
>   - In the level of algorithm, the motivation and process of "multiple generation" should be VERY clear: Monte-Carlo sampling and LLM generation are common and familiar to the community. Thus, we omit some details for compactness.
>   - In the level of implementation, we actually introduce the generation details in line 937-944 in appendix C, including the prompt and how we maintain the template of the generation. We also present some examples in Table 2. Reviewer Vz7F can check this for details.
>
> **Reasons to Reject#2:** Time complexity concerns.
>
> - **We believe that Reviewer Vz7F has a misunderstanding about the paper**. $L$ is NOT the generated sequence length, it's the number of generated sequences for estimating $p(y)$. According to line 941-944, the generation ends when the line break "\n" is sampled or the generation length exceeds to 384 (The actual length is generally far smaller than 384). According to appendix I, the calibration results converge rapidly as $L$ increases. So in our experiments, we set $L$ to just 100, which we believe is small. In deployment, if we have $N$ query examples, the method includes $L$ generations and $N$ ICL inferences, which the time complexity is $O(L+N)$. When $N>>L$, the additional time cost of the generation can be neglected. Other calibration methods except NC have the same time complexity as ours. Time complexities of the different methods are as follows:
>
>   | Method         | Complexity  | Remark                                           |
>   | -------------- | ----------- | ------------------------------------------------ |
>   | ICL            | $O(N)$      |                                                  |
>   | NC             | $O(\|Y\|N)$   | $\|Y\|$: number of labels                          |
>   | PC             | $O(L+N)$    | $L$: number of generations                       |
>   | CC             | $O(L+N)$    | $L$: number of context-free inputs               |
>   | DC             | $O(L+N)$    | $L$: number of domain context-free inputs        |
>   | LocalE/GlobalE | $O(K!L+N)$  | $L$: number of generations per-order permutation |
>   | KATE           | $O(M+N+MN)$ | $M$: size of the whole training set            |
>   | GC (Ours)      | $O(L+N)$    | $L$: number of generations                       |
>
>   **Therefore, our approach is lightweight and has no complexity concern.** We think analyzing the time complexity is straightforward and easy, so we do not elaborate it in details in the paper. We will add this table in the appendix for reference.
>
> **Question#1 & Question#2:**
>
> - For question#1, please refer to appendix C and Reply to Reasons to Reject#1. For question#2, please refer to Reply to Reasons to Reject#2.
>
> **Clarification**
>
> - We believe appendix of the paper and our response should address the concerns of Reviewer Vz7F. **We also find that those concerns have nothing to do with the soundness of the paper. Given that Reviewer Vz7F thinks that our analysis is clear and also supported by experiments well and experimental results are inspiring, we sincerely think the paper's soundness is a bit underrated and hope Reviewer Vz7F can raise the score.**

---

### Official Review · Reviewer_HMv7 · 2023-08-03

**Soundness:** 3

**Excitement:**

4: Strong: This paper deepens the understanding of some phenomenon or lowers the barriers to an existing research direction.

**Paper Topic And Main Contributions:**

The authors proposed a novel calibration based approach for the in-context classifier p(y|x) by simply adjusting the label marginal p(y), which can be obtained by marginalizing out the input x via Monte-Carlo sampling. The results show that the proposed approach greatly and consistently improves the baselines for different LLM scales with stability.

**Questions For The Authors:**

Q1) What is the relationship between the topic prior p(θ) in LDA and that used in LLM? It is still confusing. Q2) What is the definition of the topic? (the Multinomial distribution on words?)

**Reasons To Accept:**

1. The method is a calibration method which is lightweight for reducing the the instability.
2. It provides a new viewpoint from the Bayesian interpretation perspective of ICL.
3. The paper explicitly validated the soundness of the previous assumption, i.e., the ICL predictive distribution has a shift on label marginal.
4. Exhaustive experiments on 12 text classification tasks and 12 LLMs scaling from 774M to 33B are provided.

**Reasons To Reject:**

1. The key assumption that "the in-context label marginal p(y) generally deviates from the true one q(y)" is only justified by empirical evidences in Section 3.2, without theoretical proofs.
2. The Equations (1) and (2) are not very readble.
3. The topic prior p(θ) is not well discussed for interpreting the meaning for LLM.

**Reproducibility:**

4: Could mostly reproduce the results, but there may be some variation because of sample variance or minor variations in their interpretation of the protocol or method.

**Reviewer Confidence:**

3: Pretty sure, but there's a chance I missed something. Although I have a good feel for this area in general, I did not carefully check the paper's details, e.g., the math, experimental design, or novelty.

---

> ### Author Rebuttal · Authors · 2023-08-29
>
> We sincerely thank Reviewer HMv7 for the valuable feedback. Now we will reply one by one according to the points mentioned by the reviewers.
>
> **Reasons to Reject#1:** The key finding is only justified by empirical evidences without theoretical proofs.
>
> - **We strongly disagree to take this as a reason to reject** for the following reasons:
>   - We actually have theoretical analysis in Section 3.1, which we think is clear and enough for illustration. Our empirical evidence is also sufficient in Section 3.2, which Reviewer HMv7 has also pointed out as a reason to accept.
>   - "The in-context label marginal $p(y)$ generally deviates from the true one $q(y)$" is NOT an assumption, but a key finding in our paper. For empirical findings, we believe there is no mandatory requirement for theoretical proofs, especially given that recent popular works like CoT hardly have theoretical support but also greatly promote the development of NLP. Thus, we feel this criticism is a bit unfair.
>   - We think this criticism falls into the quick heuristics "The paper doesn't use [my preferred methodology], e.g., deep learning" listed in the review policy of EMNLP2023. NLP is an an interdisciplinary field and relies on many kinds of contributions including data/model analysis and theory. There should be no distinction between different types of contributions, as long as it is enough for illustration.
>   - The goal of the paper is to calibrate the ICL prediction, not theoretical proofs of why the shift happens.
>
> **Reasons to Reject#2:** Equations (1) and (2) are not very readable.
> - **We strongly disagree to take this as the reason to reject.** Equation (1) and (2) require those notations for mathematical rigor and self-contained, and it's also readable enough given the background. Perhaps there is a more compact way for expression and we will revise this, this can't be so strong to be a reason to reject.
>
> **Reasons to Reject#3:** The topic prior $p(\theta)$ is not well discussed.
> - Topic refers to the task as shown in footnote 2, and we just follow the convention to call it "topic". The topic prior $p(\theta)$ is the marginal distribution of the topic in the pretrained corpus. We can simply take it as the topic's occurrence frequency in the pretrained corpus since the latter is it's MLE estimation. **While topic prior $p(\theta)$ is detailedly discussed in the previous work in ICL's Bayesian interpretation like [1], we don't talk about it too much but only discuss its effect to ICL in line 175-182. We think this discussion is proper and enough. We apologize for the non-clearness and will add more details about the Bayesian interpretation in the appendix.**
>
> **Question#1:** Relationship between the topic prior p(θ) in LDA and that used in LLM
> - To answer this question, we first clarify the the relation and difference between LDA and LLMs:
>   - Relation: Both LDA and LLMs are to model the text distribution $p(x)$.
>   - Difference: With different goals, LDA and LLMs have completely different assumptions and practices.
>     - LDA aims to cluster the words in the document, then it has a latent topic assumption: a document is generated by first sampling topic $\theta$ from topic prior $p(\theta)$, and then sampling a document from $p(x|\theta)$. Then, the text distribution $p(x)$ is obtained via marginalization: $p(x)=\int p(x|\theta)p(\theta)d\theta$. In LDA, $\theta$ is a Dirichlet variable lies in a simplex $\Delta^{P-1}$, where $P$ is the number of the predefined word clusters (There are other latent variables in LDA, but we only talk about the topmost one for compactness). Hence, each element in $\theta$ represents the probability that the word in the corresponding cluster appears in the document. Note that this assumption greatly simplifies dependencies in text (for example, it considers the document as a bag-of-word). However, the goal is not generation but to use the posterior $p(\theta|x)$ for clustering. Thus, this assumption is reasonable.
>     - LLMs aim to generate fluent and coherent sentences, then they model the text distribution $p(x)$ directly. In specific, LLMs consider $p(x)$ to be a cascade of Categorical distributions, which can capture any dependencies in text in theory.
>
>   So how do we connect LDA and LLMs? The logic is that though LLMs directly model the text distribution $p(x)$, the community generally consider the text, i.e., the pretrained corpus, is structured and arranged by some latent variables controling the generation of text like the assumption in LDA. Therefore, if a LLM $p(x)$ approaches the true text distribution exactly (which is a widely used assumption in theoretical analysis and first proposed in [1]), $p(x)$ can be decomposed into the marginalization of these latent variables.
>
>   Here, the key problem is that we don't know what exactly the latent variables as well as the generative structure look like. Previous works have different but also self-contained assumptions such as mixture of HMMs [1], constituency tree [2], or even embedding [3]. Basically, we can call all of these latent variables as "topic", since they abstract the content of text and then play a role of clustering.
>
>   **In our work, we just assume the topic existence but not the form: we presume there is a topic $\theta$ existing in the pretrained corpus such that $p(x,y|\theta)$ can model the distribution of examples in the target task, and we doesn't assume the specific mathematical form of the topic. This assumption is widely used in previous works and also enough for our analysis.** So we apologize for the  confusion and we will add more clarifications to make the paper be more clear.
>
> **Question#2:** What is the definition of the topic?
>
> - According to reply to Question#1, topic is only a latent variable abstracting the task and we don't care about the specific form. However, the assumption of multinomial distribution on words as Reviewer HMv7 mentioned is non-proper, since it's a bag-of-word assumption but clearly the task example has dependencies between input sequences and output label tokens.
>
> **Clarifications**
>
> - We note that Reviewer HMv7 has much concerns about the relation between topic models and LLMs. While the connection is detailedly discussed in the previous works [1][2][3], our paper only takes it as a tool for theoretical analysis, which is for scientific rigor and not the main contribution of our works. For deeper understanding on this topic, Reviewer HMv7 can refer to those previous works. Because of this and considering the contribution of our work, we sincerely think the paper's soundness is a bit underrated and hope Reviewer HMv7 can raise the score.
>
> **References**
>
> 1. Xie S M, Raghunathan A, Liang P, et al. An explanation of in-context learning as implicit bayesian inference[J]. arXiv preprint arXiv:2111.02080, 2021.
>
> 2. Hahn M, Goyal N. A theory of emergent in-context learning as implicit structure induction[J]. arXiv preprint arXiv:2303.07971, 2023.
>
> 3. Wang X, Zhu W, Wang W Y. Large language models are implicitly topic models: Explaining and finding good demonstrations for in-context learning[J]. arXiv preprint arXiv:2301.11916, 2023.

---

### Meta-Review · Area_Chair_nCoC · 2023-09-23

**Recommendation:** 3

**Metareview:**

The reviewers are agreed on the excitement, but are finding small errors and issues that need further clarification in the text. These should be cleared up in further drafts of this work.

---

### Decision · Program_Chairs · 2023-10-07

**Decision:**

Accept-Findings

**Comment:**

The reviewers are agreed on the excitement, but are finding small errors and issues that need further clarification in the text. These should be cleared up in further drafts of this work.